# Copper to Tuscany – Coals to Newcastle? The dynamics of metalwork exchange in early Italy

Andrea Dolfini[1]☯*, Ivana Angelini[2]☯, Gilberto Artioli[2]☯

**1** School of History, Classics and Archaeology, Newcastle University, Newcastle upon Tyne, England, United Kingdom, **2** Department of Geosciences, University of Padua, Padova, Italy

☯ These authors contributed equally to this work.
* andrea.dolfini@ncl.ac.uk

**Data Availability Statement:** All relevant data are within the manuscript and its Supporting Information files. In particular, the archaeological data are laid out in Table 2, the Pb isotope data are

## Abstract

The paper discusses results of an interdisciplinary research project integrating lead isotope, chemical, and archaeological analysis of 20 early metal objects from central Italy. The aim of the research was to develop robust provenance hypotheses for 4th and 3rd millennia BC metals from an important, yet hitherto neglected, metallurgical district in prehistoric Europe, displaying precocious copper mining and smelting, as well as socially significant uses of metals in 'Rinaldone-style' burials. All major (and most minor) ore bodies from Tuscany and neighbouring regions were characterised chemically and isotopically, and 20 Copper Age axe-heads, daggers and halberds were sampled and analysed. The objects were also reassessed archaeologically, paying special attention to find context, typology, and chronology. This multi-pronged approach has allowed us to challenge received wisdom concerning the local character of early metal production and exchange in the region. The research has shown that most objects were likely manufactured in west-central Italy using copper from Southern Tuscany and, quite possibly, the Apuanian Alps. A few objects, however, display isotopic and chemical signatures compatible with the Western Alpine and, in one case, French ore deposits. This shows that the Copper Age communities of west-central Italy participated in superregional exchange networks tying together the middle/upper Tyrrhenian region, the western Alps, and perhaps the French Midi. These networks were largely independent from other metal displacement circuits in operation at the time, which embraced the north-Alpine region and the south-eastern Alps, respectively.

## Introduction

The last two decades have witnessed a major surge of interest in the origins of copper mining, smelting, and working in the Italian peninsula. The new research season was inaugurated by fieldwork and radiocarbon dating at Libiola and Monte Loreto, two copper mines from eastern Liguria (Fig 1). Investigations brought to light prehistoric workings and galleries for the extraction of chalcopyrite (presumably supplemented by near-surface deposits of copper oxides/carbonates) dating from the mid-4th millennium BC. This pushed back significantly the beginnings of copper mining south of the Alps [1–2]. Such a surprising discovery led to

laid out in Table 4, and the chemical composition data are laid out in Table 5.

**Funding:** Chemical and isotopic analyses of ore and object samples were funded by the University of Padua, who also covered all travel expenses for Angelini. Dolfini's research trips were funded by the Historical Metallurgy Society, Coghlan Bequest Fund.

**Competing interests:** The authors have declared that no competing interests exist.

reconsidering the chronology of early Italian metals, which at the time were overwhelmingly dated to the 3rd millennium BC [3–4]. A review of the evidence showed instead that Italian copper production likely commenced in the late 5th or early 4th millennium BC (i.e. the late/final Neolithic). It also showed that Neolithic metalworkers did not just manufacture small awls and points, as was until then believed, but also large, technologically complex axe-heads [5]. The discovery of final Neolithic smelting evidence from Orti Bottagone, Tuscany, further corroborated the new picture, showing that copper was not only cast and worked, but also smelted (and presumably mined) in the area from the early 4th millennium BC [6–7:p.147].

Further research indicated that metal production and use surged dramatically in the early Copper Age, 3600–3350 BC. In the ore districts of Tuscany, in particular, the mid-4th millennium BC marked the inception of a mature technological tradition that, somewhat inappropriately, has come to be known as 'Rinaldone metallurgy' after a central Italian burial custom [8]. So-called 'Rinaldone metallurgy' featured: (a) the mastering of copper and arsenical-copper alloys (including a ternary Cu-As-Sb alloy that is typically found in central Italy) [9–10]; (b) early experimentation with silver and antimony extraction and casting; (c) the manufacture of distinctive objects including personal ornaments, axe-heads, daggers, and halberds; (d) technological improvements in metal casting and working; and (e) important changes in the cultural signification of metals, which entered the funerary domain to mark new ideas of gender, age, and perhaps status [11–13]. Such uses of metal objects are most apparent in iconic 'Rinaldone' warrior burials [8,14–15].

More recently, slag analysis from the now-destroyed settlement site of San Carlo-Cava Solvay has enabled further insights into early extractive metallurgy. Nestled at the edge of the Tuscan Colline Metallifere (or 'Ore Mountains'), the site was excavated in the 1990s under rescue conditions [7]. The fieldwork brought to light several fired-clay platforms partly surrounded by curvilineal limestone walls, similar to early smelting installations from the eastern Alps and southern France [16–18]. Several copper-based compounds including sulphides and sulphosalts were smelted at the site using a surprisingly efficient reduction technology. This seemingly involved the crucible smelting of mixed charges of copper sulphides and oxides/carbonates, aided by blowpipes and/or bellows [19]. The process generated mature silica slags lacking the unreacted ore and entrapped copper prills typically found in Chalcolithic metallurgical residues [20]. This is all the more remarkable considering that the site dates to the late 4th millennium BC [7:p.142]; this makes it one of the earliest metallurgical sites in Western Europe.

Despite these advances, a crucial field of research has remained woefully underdeveloped in Italian archaeometallurgy, namely the study of early metalwork exchange. The topic was recently addressed for the Alpine region using, alternatively, lead isotope analysis (LIA) [21–22] and the so-called 'Oxford system' charting changes in trace element composition and alloy type over time and geography [23–24]. South of this area, however, the provenance and circulation dynamics of early metals have never been researched on a meaningful scale. Consequently, fundamental questions concerning the role of metals in the major social transformations affecting the Italian peninsula in the 4th and 3rd millennia BC are still unanswered [25]. Some are surprisingly basic, *viz.* was all central Italian metalwork fashioned from local copper? How far did metal objects travel from the ore districts of Tuscany and Liguria? And was alien copper imported into the region? Answering these questions would enable us to interrogate the record with increasingly sophisticated queries regarding exchange modes and mechanisms. Was metal primarily circulated over short distances, perhaps by direct exchange? Or did it travel further afield through multiple handovers? If so, what was the reach of the network? Did early metals 'make the world go round', as Pare [26] and many others suggest they did in the Bronze Age, or did they perform more restricted social functions prior to this time?

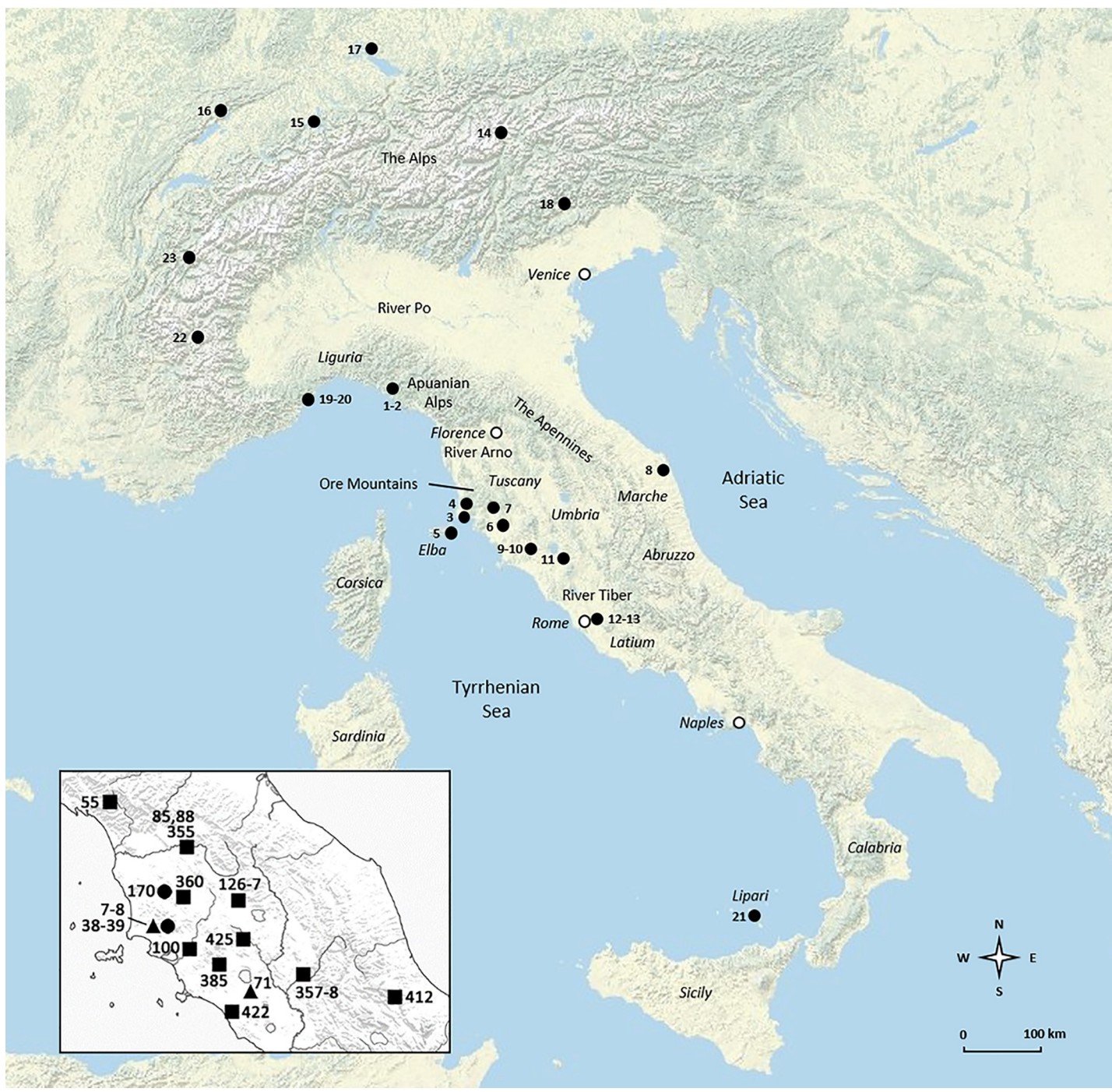

**Fig 1. Map of the sites mentioned in the article. Inset: find spots of the metal objects analysed.** 1: Libiola; 2: Monte Loreto; 3: Orti Bottagone; 4: San Carlo-Cava Solvay; 5: Grotta San Giuseppe; 6: Grotta del Fontino; 7: Grotta della Spinosa; 8: Fontenoce di Recanati; 9: Ponte San Pietro; 10: Selvicciola; 11: Rinaldone; 12: Lucrezia Romana; 13: Romanina; 14: The Iceman; 15: Zug-Riedmatt; 16: Saint Blaise/Bains des Dames; 17: Hornstaad; 18: Col del Buson; 19: Arene Candide; 20: Grotta della Pollera; 21: Lipari; 22: Saint Véran; 23: Fontaine-le-Puits. Hollow circles: contemporary cities. Inset: squares: axe-heads; dots: daggers; triangles: halberds. Objects ID 7–8, 38–39 (plus hafting rivet ID 39A): Pianizzoli; 55: Garfagnana; 71: Vetralla; 85, 88, 355: Province of Florence; 100: Province of Grosseto; 126–127: Il Teso; 170: Querceto; 357–358: Near Terni; 360: Province of Siena; 385: Selvena; 412: Abruzzo; 422: Corneto (modern-day Tarquinia); 425: Sarteano, Palazzone. Note that the locations of objects ID 55, 85, 88, 100, 355, 357, 358, 360 and 412 are approximate (image: Andrea Dolfini; base map: U.S. Geological Survey).

This article provides a science-based, and theoretically informed, pathway towards addressing these questions. Its aim is to assess metalwork circulation and exchange in 4[th] and 3[rd] millennia BC central Italy through the critical application of LIA [27–29]. Aside from sporadic attempts based on very few samples [30–31], this analytical method has not yet been applied to this region. Importantly, the isotopic data will be cross-referenced with, and augmented by, archaeological and copper chemistry datasets, which are indispensable for testing the provenance hypotheses generated by LIA [22,28–29,32]. As will be shown below, this multi-pronged approach has yielded original insights into the social dynamics of copper procurement and displacement in one of the most important mining and metalworking districts of early Europe.

## Cultural context

In the late and final Neolithic (4500–3600 BC), Italy partook in the wide-ranging communication networks and distinctive social phenomena characterising much of the central Mediterranean region (Table 1). These encompassed the long-distance exchange of obsidian from Lipari and other island sources; the emergence of a more mobile economic regime and lifestyle, which saw the abandonment of early/middle Neolithic nucleated villages and the transformation of the inhabited landscape towards smaller and shorter-lived settlements; the birth of formalised burial practices manifesting themselves in new funerary structures (e.g. rock-cut tombs) and mortuary customs (e.g. increasingly elaborate bone manipulation rituals); and a suite of technological innovations including the extractive metallurgy of copper [25,33–34]. Furthermore, new techniques of flint pressure-flaking were introduced at this time for manufacturing tanged-and-barbed arrowheads and daggers, and new stone materials such as jasper and steatite (also known as soapstone) replaced long-revered 'greenstone' and obsidian. Finally, the horse, plough, and wheel were introduced into the peninsula in the late 4[th] millennium BC, increasing agricultural output and perhaps triggering population growth [25,35].

Pottery styles provide one of the best indicators for changing communication networks from the late Neolithic to the Copper Age. In the former period, central Italy featured a marked east-west split in its cultural manifestations. East of the Apennines, late Neolithic potters fashioned bowls and plates in the Late Ripoli style, which is characterised by either plain or burnished surfaces. West of the range, however, craftspeople shaped and decorated their wares based on original combinations of the eastern Late Ripoli, north-western Chassey-Lagozza, and southern Diana ceramic styles [36–39] (Fig 2). This highlights the role of west-central Italy as a crossroads connecting north-western Italy, the mid-Adriatic coast, and the southern peninsula. Significantly, of all major ceramic styles documented in late Neolithic Italy, north-eastern Square-Mouthed Pottery, phase III (SMP III), is the sole not to be found in the region. This provides important evidence regarding the lack of communication networks linking north-east and west-central Italy. As we shall see below, this bears important implications for the unfolding of metalwork exchange in the Copper Age.

**Table 1. Chronology of prehistoric Italy, 4500–2000 BC.**

| Archaeological phase | Absolute chronology |
|---|---|
| Late Neolithic | 4500–3800 BC |
| Final Neolithic | 3800–3600 BC |
| Early Copper Age | 3600–3350 BC |
| Middle Copper Age | 3350–2800 BC |
| Late Copper Age | 2800–2200 BC |
| Early Bronze Age, phase 1 | 2200–2000 BC |

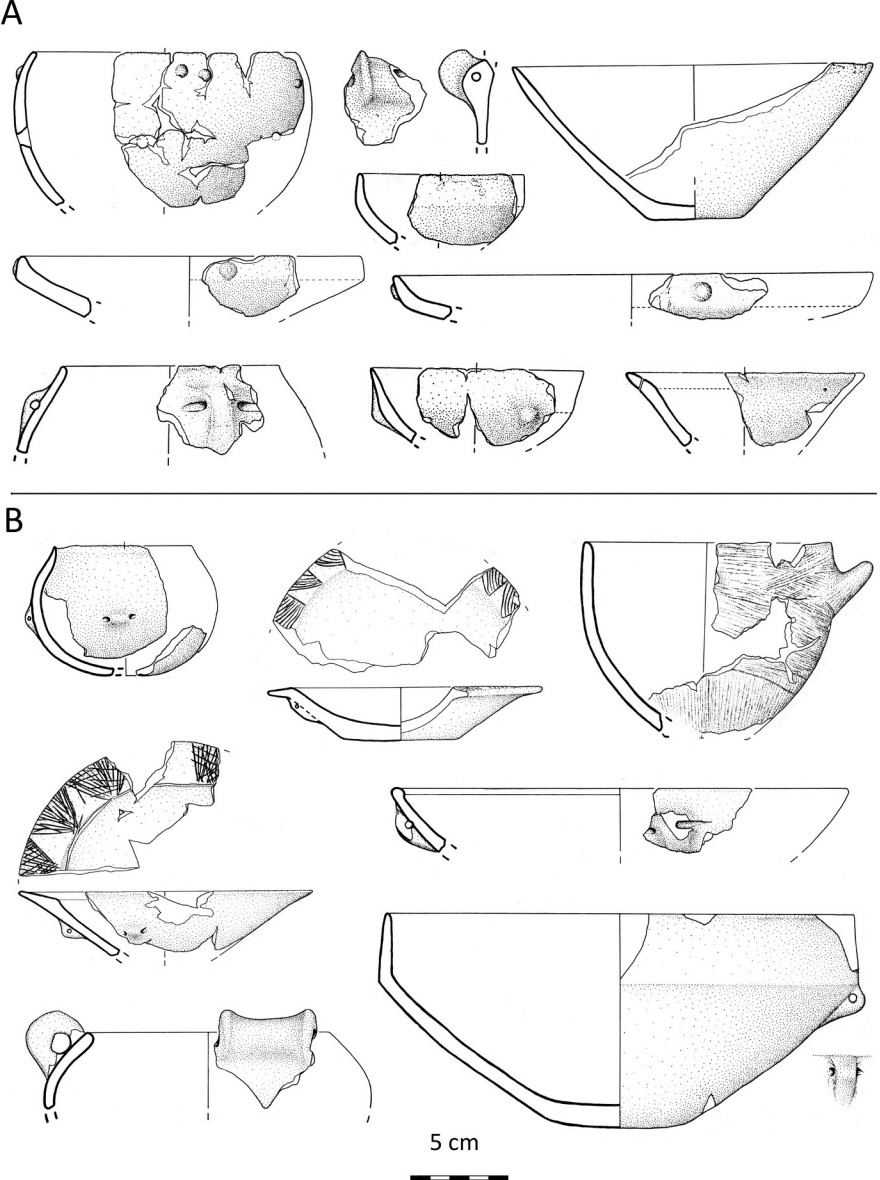

**Fig 2. Late Neolithic ceramic styles from west-central Italy.** A: Late Ripoli pottery from Casale di Valleranello, Rome. B: Pottery from Quadrato di Torre Spaccata, Rome; this is mostly in the northern Chassey-Lagozza style, but the bottom left vessel has a rocchetto (spool-shaped) handle typical of southern Diana wares (image: Giovanni Carboni).

The mid-4th millennium BC marked a 'perfect split' between domestic and funerary pottery production on both sides of the peninsula. On the one hand, the late Neolithic predilection for glossy surfaces was carried into the realm of mortuary practices, which acquired unprecedented social centrality in the Copper Age [34,40–41]. On the other hand, the taste for coarse impasto wares with textured surfaces, which had gained momentum in the final Neolithic, continued to thrive at open-air villages and other habitation sites. Extending previously localised trends to the entire central peninsula, Copper Age domestic ceramics were shaped and ornamented according to a bewildering array of stylistic traits, which household and village potters blended into idiosyncratic material assemblages [42]. As a counterpoint to the pulverisation of domestic pottery styles, superregional funerary styles emerged at this time

throughout Italy. The process is mirrored by the burial customs themselves, which, from the mid-4th millennium BC, increasingly shared common traits over wide areas. Two such customs emerged in central Italy: interment in caves, or 'Vecchiano' tradition, and burial in small hypogeal chamber graves (occasionally supplemented by trench and cist tombs), or 'Rinaldone' tradition. The former is mainly found along the Tyrrhenian littoral from eastern Liguria to southern Tuscany, while the latter clusters at three principal foci in southern Tuscany/ northern Latium, the lower Tiber valley, and the Adriatic lowlands [43–47].

In burial caves, mortuary rites centred on the inhumation of articulated bodies followed by bone manipulation and fragmentation, once the flesh had naturally decayed. The archaeological outcome of such processes are thick deposits in which large amounts of bone fragments are mingled with potsherds, dismembered ornaments, and (mostly unbroken) flint and metal weapons. Articulated bodies are all but absent. Copper-alloy daggers and halberds were deposited with relatively high frequency at these sites, along with (necklace?) beads made from metallic silver and antimony, and also steatite and animal bone [48–49]. Metal axes, on the other hand, are relatively rare, as they were preferentially deposited in the open landscape [50]. Burial caves such as Grotta del Fontino, Grotta della Spinosa, and Grotta San Giuseppe provide prime examples of this funerary behaviour [51–54].

The Rinaldone burial tradition is typified by small hypogeal cemeteries comprising clusters of rock-cut chambers accessed through short shafts or entrance corridors. The tombs normally hosted small numbers of bodies, both articulated and reorganised. Stereotyped grave sets were used to mark out gender, age, and other aspects of personal identity [34,40,55]. Typically, adult males were given copper-alloy daggers and halberds (along with flint and hardstone weapons), while women and children were buried with silver and antimony beads, as well as non-metal ornaments [56–57]. Copper awls were placed with all gender and age categories, and so were flask-shaped funerary vessels [14]. Numerous Rinaldone-style burial sites are known in central Italy. Among the most representative of the funerary behaviour described above are Fontenoce di Recanati in Adriatic Italy and Ponte San Pietro (Fig 3), Selvicciola, Rinaldone, Lucrezia Romana, and Romanina in Tyrrhenian Italy [8,43,58–65].

## Materials and methods

### Sample selection

Twenty copper-alloy objects from central Italy were selected for this study including 14 axe-heads, three daggers, two halberds, and a unique implement preliminarily interpreted here as an early halberd blade. A hafting rivet from one of the halberds was also analysed, bringing the total sample to 21 items (Table 2). All objects are securely dated to the Copper Age (3600–2200 BC) by multiple markers including typology, alloy composition, and find context. They belong to a broader selection of early metals investigated macro- and microscopically for their manufacturing technology, uses, and post-depositional histories [66]. They were also characterised in terms of chemical composition and metallography, but these data are not discussed here. Most objects in our sample had previously been analysed for their bulk alloy composition by the University of Stuttgart, the British Museum, and other research institutions [67–70]. In many cases, we deliberately targeted metals chemically characterised by others as we intended to cross-check our compositional data with those obtained by other laboratories using dissimilar analytical techniques and sample preparation protocols. Moreover, we wanted to cross-reference lead isotope (LI) signatures with trace element data, which provide complementary evidence strengthening or refuting provenance hypotheses [28–29]. All necessary permits were obtained to sample and analyse these objects, which complied with all relevant regulations. In particular, the then Soprintendenza per i Beni Archeologici della Toscana granted

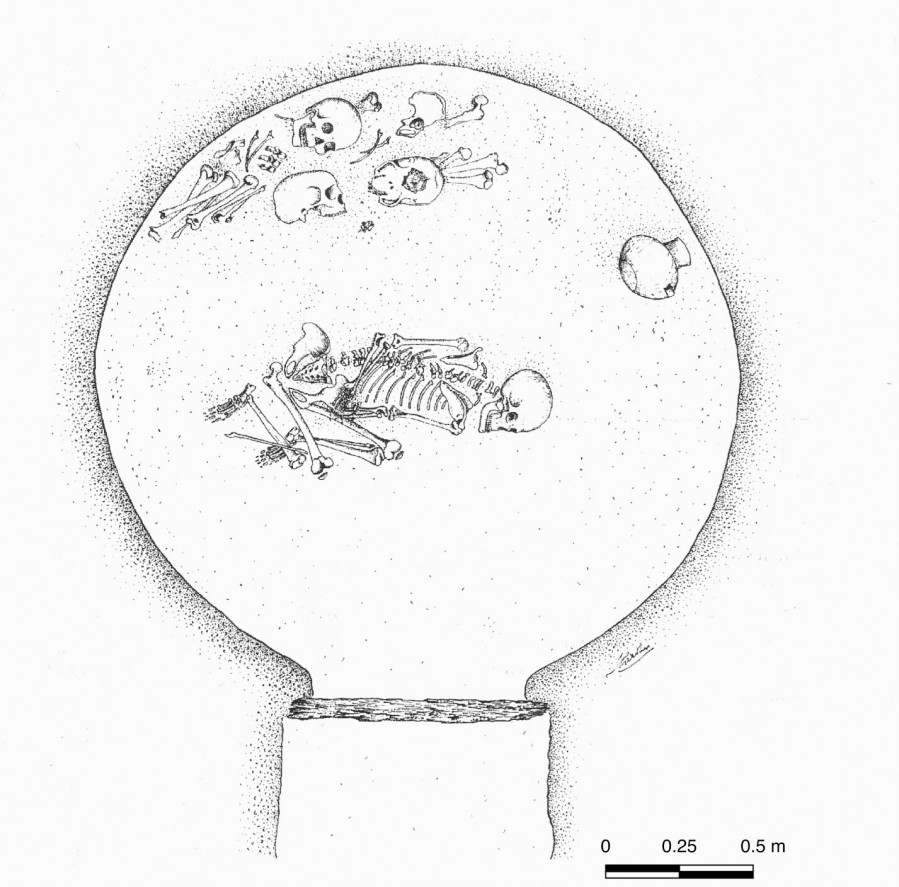

**Fig 3. Articulated burial and dismembered human remains from Ponte San Pietro, tomb 22.** The chamber tomb is typical of the Rinaldone burial custom, central Italy, *c*.3600-2200 BC. Reprinted from Miari 1995 under a CC BY licence, with permission from Monica Miari, original copyright 1995.

access to the objects kept in the Archaeological Museums of Florence, Siena and Arezzo, while the Trustees of the British Museum issued permits regarding the objects in their care.

Thirteen (out of 14) axe-heads in our sample are typologically related. They belong to a broad class of trapezoidal or sub-rectangular tools with curved (or mildly splayed) cutting edges and flat margins that may or may not have been raised by hammering (Fig 4). While the hammering of margins was until recently believed to mark a more advanced, and chronologically posterior, horizon in early Italian metallurgy, recent research has shown this to be a false proposition, for raised margins are found on 4[th] millennium as well as 3[rd] millennium BC tools [71]. This is best demonstrated by the late 4[th] millennium BC Iceman's axe-head, which features slightly raised margins [72]. The one typological outlier in the sample is a stray find from Garfagnana, northwest Tuscany (Fig 4:55). This is an unusually wide and squat axe-head with a markedly splayed cutting edge for which no close parallels are found in the Italian record.

All axe-heads were found west of the Apennines, except for a single specimen from Abruzzo, east-central Italy (Fig 1, inset:412). As is typical of Copper Age Italy, they are mostly stray finds, suggesting intentional deposition in the open landscape [50]. Two of them, however, are from a burial site. This is an atypical Rinaldone-style trench grave from Il Teso near Arezzo (Fig 1, inset:126–127) [73:p.50]. Early Italian axe-heads are often difficult to date due

**Table 2. Prehistoric metal objects analysed for this project.**

| DB ID | Lab ID | Site | Commune | Province | Region | Object | Context | Chronology | Dates BC | Museum | Accession number | Bibliography |
|---|---|---|---|---|---|---|---|---|---|---|---|---|
| 7 | Pz-Pg0 | Pianizzoli | Massa Marittima | Grosseto | Tuscany | dagger | Burial cave? | Middle Copper Age | 3350–2800 | Florence | 92480 | Bianco Peroni 1994, 137 |
| 8 | Pz-Pg2 | Pianizzoli | Massa Marittima | Grosseto | Tuscany | dagger | Burial cave? | Middle Copper Age | 3350–2800 | Florence | 92482 | Bianco Peroni 1994, 141 |
| 38 | Pz-Al9 | Pianizzoli | Massa Marittima | Grosseto | Tuscany | halberd | Burial cave? | Middle/Late Copper Age | 3350–2200 | Florence | 92479 | Bianco Peroni 1994, 272 |
| 39 | Pz-Al8 | Pianizzoli | Massa Marittima | Grosseto | Tuscany | halberd | Burial cave? | Middle/Late Copper Age | 3350–2200 | Florence | 92478 | Bianco Peroni 1994, 271 |
| 39A | Pz-R3 | Pianizzoli | Massa Marittima | Grosseto | Tuscany | rivet of ID 39 | Burial cave? | Middle/Late Copper Age | 3350–2200 | Florence | N/A | N/A |
| 55 | Gar-Axg | Garfagnana | N/A | Lucca | Tuscany | axe | N/A | Copper Age | 3600–2200 | Florence | 763 | Carancini 1993 fig.3.24 |
| 71 | Vet-Al | Vetralla | Vetralla | Viterbo | Lazio | halberd? | N/A | Early Copper Age? | 3600–3350? | Florence | 78793 | Junghans et al. 1974, 20599 |
| 85 | Fi-Ax9 | Province of Florence | N/A | Florence | Tuscany | axe | N/A | Copper Age | 3600–2200 | Florence | 1069 | Junghans et al. 1960, 597 |
| 88 | Fi-Ax1 | Province of Florence | N/A | Florence | Tuscany | axe | N/A | Early/Middle Copper Age | 3600–2800 | Florence | 1071 | Junghans et al. 1960, 594 |
| 100 | PGr-Ax | Province of Grosseto | N/A | Grosseto | Tuscany | axe | N/A | Copper Age | 3600–2200 | Siena | N/A | Junghans et al. 1960, 606 |
| 126 | Te7-Ax | Il Teso | Marciano della Chiana | Arezzo | Tuscany | axe | Trench grave | Early Copper Age | 3600–3350 | Arezzo | 967 | Cocchi & Grifoni 1989, fig.22A.1 |
| 127 | Te8-Ax | Il Teso | Marciano della Chiana | Arezzo | Tuscany | axe | Trench grave | Early Copper Age | 3600–3350 | Arezzo | 968 | Cocchi & Grifoni 1989 fig.22A.4 Carancini 1993, p.129, 6 |
| 170 | Que-Pg | Querceto | Casole d'Elsa | Siena | Tuscany | dagger | N/A | Early/Middle Copper Age | 3600–2800 | Siena | N/A | Bianco Peroni 1994, 116 |
| 355 | Fi-Ax0 | Province of Florence | N/A | Florence | Tuscany | axe | N/A | Copper Age | 3600–2200 | Florence | 1030 | Junghans et al. 1960, 596 |
| 357 | BM-00 | Near Terni | Terni? | Terni | Umbria | axe | N/A | Early/Middle Copper Age | 3600–2800 | British Museum | 1964.12.1-200(288) | Bietti Sestieri & Macnamara 2007, 15 |
| 358 | BM-01 | Near Terni | Terni? | Terni | Umbria | axe | N/A | Early Copper Age | 3600–3350 | British Museum | 1964.12.1-201(300) | Bietti Sestieri & Macnamara 2007, 17 |
| 360 | PSi7-Ax | Province of Siena | N/A | Siena | Tuscany | axe | N/A | Early Copper Age | 3600–3350 | Siena | 37187 | Junghans et al. 1960, 654 |
| 385 | Sel-Ax | Selvena | Castell'Azzara | Grosseto | Tuscany | axe | N/A | Early Copper Age | 3600–3350 | Siena | 37163 | Carancini 1993, p.130, 30 |
| 412 | BM-26 | Abruzzo | N/A | N/A | Abruzzo | axe | N/A | Early Copper Age | 3600–3350 | British Museum | PRB1883.4–26.1 | Bietti Sestieri & Macnamara 2007, 4 |
| 422 | BM-47 | Corneto (Tarquinia) | Tarquinia | Viterbo | Lazio | axe | N/A | Early Copper Age | 3600–3350 | British Museum | PRB WG1047 | Bietti Sestieri & Macnamara 2007, 5 |
| 425 | Pal8-Ax | Sarteano, Palazzone | Sarteano | Siena | Tuscany | axe | N/A | Early Copper Age | 3600–3350 | Siena | 37178 | Unpublished |

Prehistoric metal objects sampled and analysed for the research. Numbers in the DB ID column (first left) refer to the database of early central Italian metals compiled by Dolfini [50,66], while the Lab ID column (second from left) lists the sample lab codes used by the University of Padua. The objects are kept in the following museum collections (Museum column, third from right): Florence: Museo Archeologico Nazionale di Firenze, Italy; Siena: Museo Archeologico Nazionale di Siena, Italy; Arezzo: Museo Archeologico Nazionale di Arezzo, Italy; British Museum: The British Museum, London, UK.

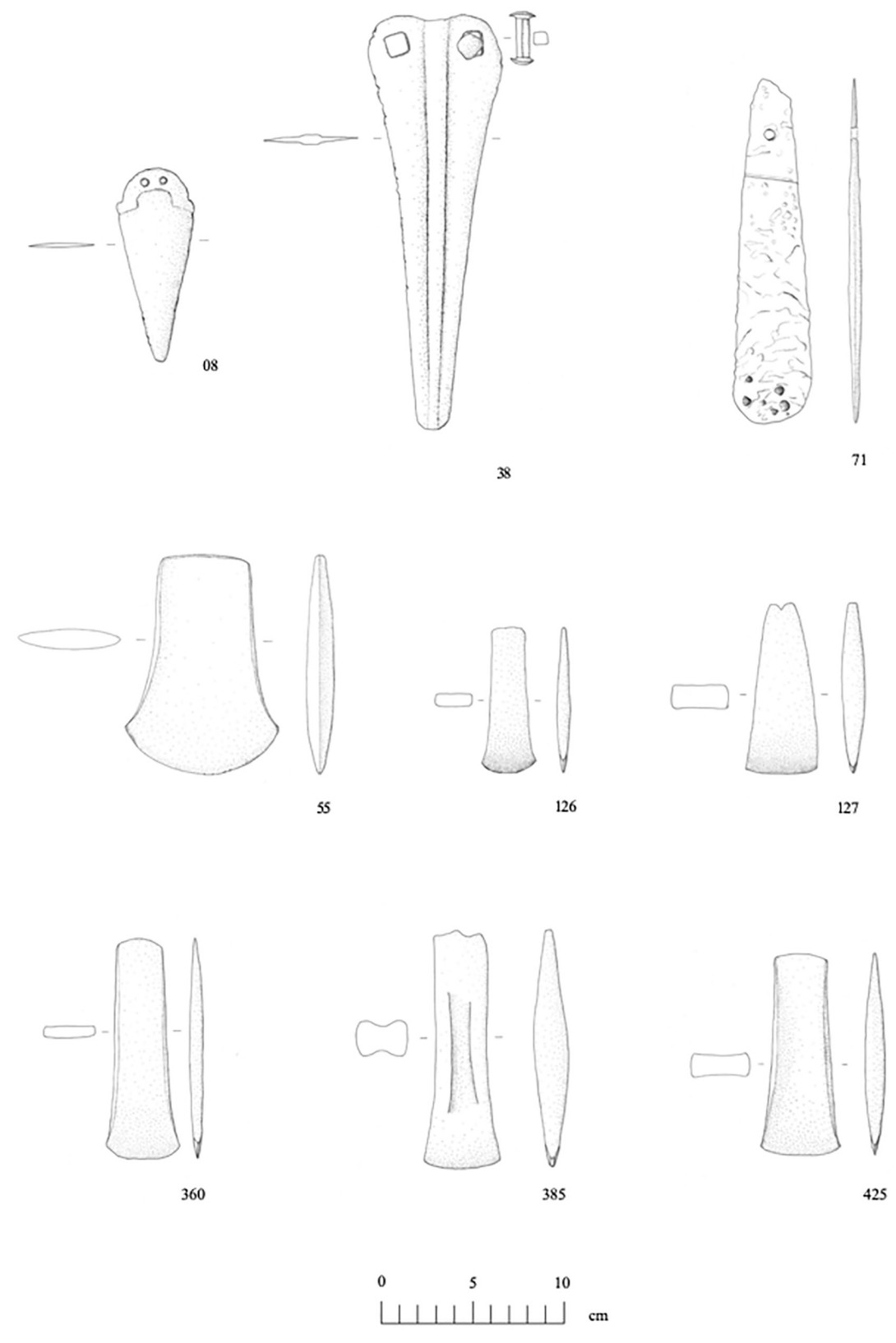

**Fig 4. Selection of copper-alloy axes, daggers and halberds analysed for the research.** ID 08: Pianizzoli (Massa Marittima type); 38: Pianizzoli (Montemerano type); 55: Garfagnana; 71: Vetralla; 126–127: Il Teso; 360: Province of Siena; 385: Selvena; 425: Sarteano, Palazzone (image: Eleonora Montanari).

to their unspecific morphologies. The chronologies proposed here are based on a recent EU-funded project aiming to precisely date early Italian metals, including those discussed here, by combined morphometric analysis, radiocarbon dating, and a critical review of find contexts [71]. Based on the project's findings, it is proposed here that ten axe-heads be dated to the early-middle Copper Age (3600–2800 BC) on typological grounds or through associated finds, the remainder being generically Copper Age (3600–2200 BC; Table 2).

Of the three daggers analysed, two are from Pianizzoli, a small burial cave in southern Tuscany, while the other is a stray find from Querceto near Siena (Fig 1, inset:7–8 and 170) [73: p.67]. The two daggers from Pianizzoli are classified under the Massa Marittima type, while the specimen from Querceto is a Guardistallo-type dagger. Implements of both types feature triangular blades without hafting tangs, but they otherwise differ in terms of butt shape, number of rivet holes, and presence/absence of a midrib. Guardistallo-type daggers were manufactured from about 3600–2800 BC, while. Massa Marittima-type daggers are dated to 3350–2800 BC [11,71; *contra* 3–4; Table 2].

Three halberds (plus a hafting rivet) were also sampled and analysed; two are from Pianizzoli–the burial cave discussed above–and one is from Vetralla. Both halberds from Pianizzoli feature elongated triangular blades with stout midribs, rounded points, and square rivet holes. They are classified under the Montemerano type, which is tentatively assigned to the mid/late Copper Age, *circa* 3350–2200 BC, based on a review of find contexts (Fig 1, inset:38–39) [9,71; *contra* 3–4; Table 2]. The Vetralla halberd is unlike any known halberd type in Europe. It consists of a flat piece of metal in the shape of an elongated teardrop, its 'business end' being the rounded, blunted drop-end edge. The blade tapers towards the butt, which is bored by a round rivet hole. Interestingly, the butt shows an oblique hafting mark, suggesting that the blade was attached to a handle or pole at right angle–hence the halberd interpretation proposed here (Fig 1, inset:71). The object was ostensibly cast in a one-piece mould [66]. It is extremely difficult to propose a chronology for this stray find due to the lack of contextual data and meaningful *comparanda*. We tentatively suggest a date in the mid-4th millennium BC, and perhaps earlier, considering the object's archaic shape and features, indicating poor command over casting technology [66]. This is consistent with what we know about the beginnings of metalworking in the Italian peninsula [5,13].

## Ore geology and lead isotope fingerprinting

Italy displays a major east-west divide in ore geology. The west-central peninsula hosts considerable ore bodies in a broad strip of land stretching from eastern Liguria in the north to upper Latium in the south, also extending into the northern Apennines up to the hinterland of Arezzo (5). Ore deposits are most conspicuous in present-day Tuscany (Fig 5); they encompass iron, base metals (including copper and localised tin), silver, antimony, mercury, and gold [74–76]. No metal-bearing sources are found east of the Apennines or south of the Tiber Valley up to Calabria in Italy's 'toe' [77–78].

The central Italian deposits have a twofold genesis. Some were born out of volcano-sedimentary, magmatic, metamorphic, and geothermal environments of pre-Tethyan and Alpine ages, while others were formed by later processes of magmatic intrusion, contact metamorphism, and hydrothermal activity [79–80]. Ore bodies of the first type derive from the ocean-related massive sulphides (VMS) embedded in the sedimentary pile during the tectonic closure

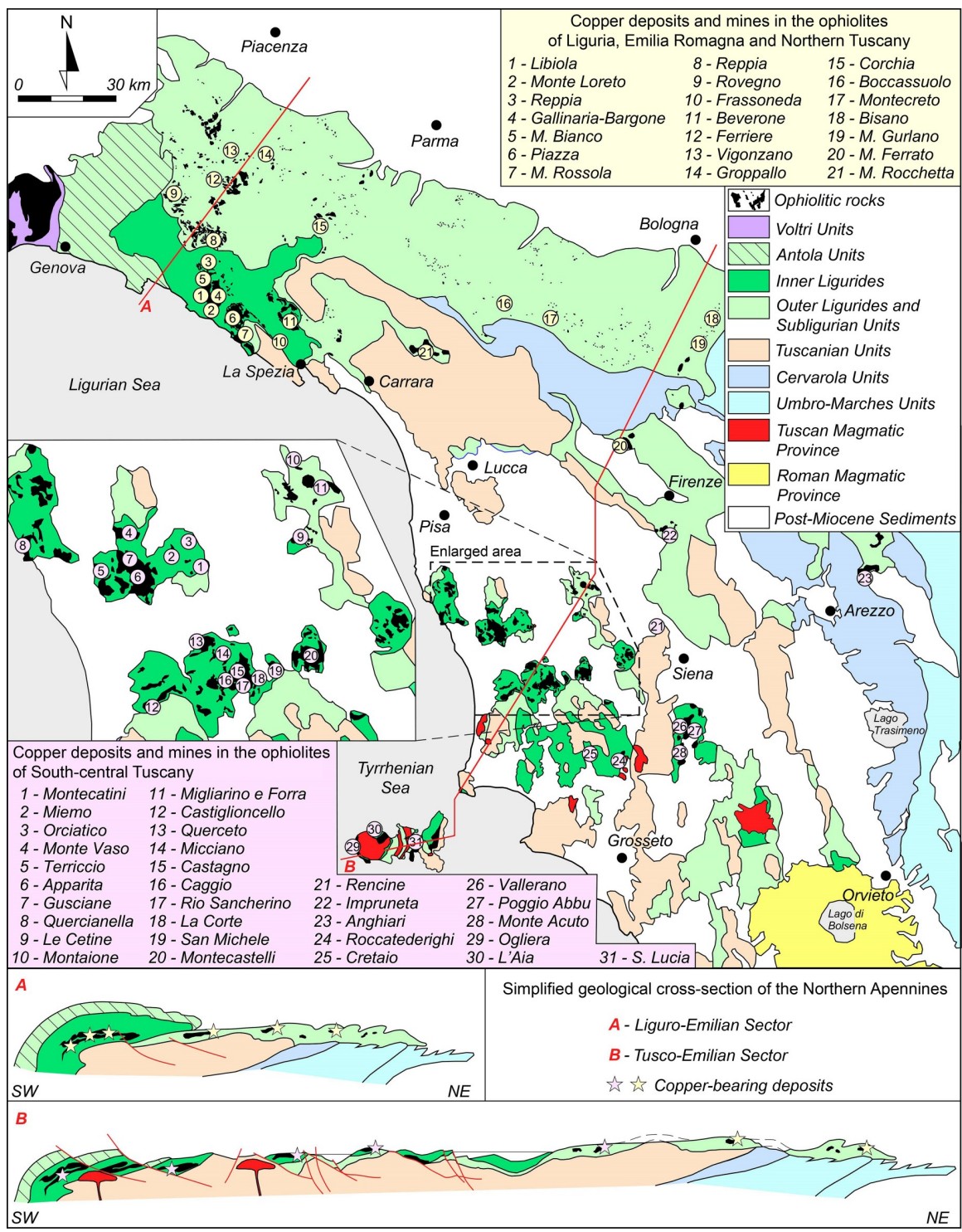

**Fig 5. Simplified geological map of west-central Italy showing the main copper deposits and mines.** Reprinted (with modifications) after Dini and Boschi 2017 under a CC BY licence, with permission from Andrea Dini, original copyright 2017.

of the western part of the Thetys [81]. These include major copper deposits hosted in the Apennine ophiolites as well as ore bodies located in central-southern Tuscany and the Elba

Island [31,82]. The central Tuscan deposits associated with ophiolitic suites originate from the complex tectonic history of the Tuscanian sector of the Northern Apennines (Tuscanian Units); they mainly consist of copper sulphides (i.e. chalcopyrite with subordinate bornite, chalcocite, and covellite), carbonates (i.e. malachite and azurite), and native copper. Iron, lead, and zinc-bearing minerals are also abundant in these geological formations but were not exploited for metallurgical purposes in the period discussed here [76,82–83]. Another suite of copper-bearing ophiolites linked to the Apennine orogeny (inner and outer Ligurides) is found along the main Apennine chain from Liguria through to inner Tuscany, and also in the Tuscan and Emilian sectors of the range. The complex tectonics acting during the Apennine orogenic compression produced a sort of 'tectonic sandwich' that caused the outer Ligurides to be embedded between the inner Ligurides and the Tuscanian Units. The most significant ophiolitic deposit is Montecatini-Val di Cecina. In the late 19th century, it generated one of the largest copper yields in the world, but there is currently no evidence that it was exploited in prehistoric times [84].

Partly overlapping with the ophiolitic deposits of oceanic origin are the ore bodies formed by later magmatic, metamorphic, and hydrothermal activity [79]. They encompass several polymetallic sulphides (Cu-Pb-Zn-Ag±Au) that are especially abundant in southern Tuscany. The strata-bound pyrite deposits of the Colline Metallifere (or 'Ore Mountains'), and in particular the Cu-Pb-Zn (±Fe, Ag, Sn) skarn rocks of the Campigliese district, are emblematic of this geology [85]. Overall, these deposits comprise a diverse suite of primary ores including, but not limited to, chalcopyrite, magnetite, sphalerite, pyrrhotite and silver-bearing galena, with minor pyrite, hematite and traces of bismuthinite, galenobismutite and many other phases [86–87]. A number of supergene copper minerals are also documented, such as cuprite, malachite, azurite, aurichalcite, brochantite, antlerite, chalcanthite, chrysocolla, and many others; native copper is present [88].

As part of this research, the LI signatures of all major, and most minor, ore deposits described above were collated from the literature and supplemented by extensive sampling campaigns coordinated by one of the authors (GA). The ore samples were characterised mineralogically and petrographically and their LI ratios were measured based on the protocols described in Artioli et al. [22]. Details of the ore bodies thus characterised and measured are provided in Table 3.

Fig 6 shows that the central Italian ore deposits are reasonably well discriminated into three major Pb isotope fields based on their relative ages and geological formation processes. They are: (a) the 'Ligurian-Apennine' field, encompassing most of the Apennine ophiolites; this mostly shows a clear mantle character with very low values of all three isotopic ratios; occasional shifts towards higher $^{207}$Pb/$^{204}$Pb, and $^{208}$Pb/$^{204}$Pb values are due to the metamorphic re-mobilisation of the metals; (b) the 'Apuanian Alps' field; and (c) the 'Southern Tuscany' field. The latter two fields have a clear metamorphic character and younger age resulting in substantially higher $^{207}$Pb/$^{204}$Pb and $^{208}$Pb/$^{204}$Pb values compared with the Ligurian-Apennine ophiolites. Consequently, the Ligurian-Apennine and Apuanian/Southern Tuscan fields can be discriminated well based on their respective isotopic ratios, reflecting ore body age and geochemical source. Areas of overlap between these ore fields and other copper deposits from Italy, Sardinia, the Alps, central Europe, and the wider Mediterranean region are discussed in Artioli et al. and Nimis et al. [21–22,89–90]. Here we remark that, by using all three LI plots, the central Italian ore deposits can be distinguished from some of the most important copper deposits in Europe including the Alps, southern France, and the central Balkans (Fig 6C). This enables the formulation of science-based provenance hypotheses for early Italian metals, bearing in mind that LI ratios are thought to determine with a high degree of certainty where the metal *could not* have come from, and that positive matches indicate the source from which the

**Table 3. Principal copper-bearing deposits of Central Italy.**

| Geological formation | Area | Main mines/ore bodies | Ore group |
|---|---|---|---|
| Inner Ligurides (Jurassic ophiolites embedded in Jurassic-Cretaceous sediments) | Ligurian Apennines | Libiola, Monte Loreto, Reppia | Ligurian Apennines |
| Outer Ligurides (Jurassic ophiolites embedded in Cretaceous turbidites) | Tusco-Emilian Apennines | Corchia, Boccassuolo, Val di Secchia, Vigonzano, Impruneta | Ligurian Apennines |
| Inner Ligurides–Tuscanian Units (Jurassic ophiolites embedded in Jurassic-Cretaceous sediments) | Ophiolites of the Tuscanian Units | Montecatini-Val di Cecina, Impruneta | Ligurian Apennines |
| Outer Ligurides–Umbro-Marches Units (Jurassic ophiolites embedded in the sediments and carbonates of the Adriatic margin) | Ophiolites of the Umbro-Marches Units | Monti Rognosi, Arezzo, Anghiari | Ligurian Apennines |
| Apuanian metamorphic unit (Triassic orogeny overprinted by Apennine orogeny with peak metamorphism 15–20 My ago) | Apuanian Alps (northwest Tuscany) | Bottino, Pollone, Monte Arsiccio | Apuanian Alps |
| Tuscanian Domain (Ophiolitic s in Mid-Tertiary Tuscan Nappe) | Colline Metallifere, Campigliese, Grossetano (central-southern Tuscany) | Boccheggiano, Campiano, Campiglia Marittima, Temperino, Lanzi | Southern Tuscany |
| Tuscanian Domain (Ophiolitic s in Mid-Tertiary Tuscan Nappe) | Elba Island | Vallone, Capo Calamita, Santa Lucia, Sant'Andrea | Southern Tuscany |
| Tuscanian Domain–Tertiary magmatism (Magmatism and metamorphism of Eocene sediments related to the plutonic complex) | Massetano (southern Tuscany) | La Pesta, Fenice Capanne | Southern Tuscany |

Principal copper-bearing deposits of central Italy, north to south. The LI signatures of all major, and most minor, ore bodies and mines listed here were either collated from the literature or newly obtained during the project.

metal *could* have originated [28–29]. In the past, neglecting this fundamental principle led to controversial, or manifestly incorrect, interpretations, and we are keen to stress that our positive matches indicate consistency with a potential source, not certain derivation. However, our provenance hypotheses are often strengthened by complementary chemical and archaeological considerations, as discussed below. When in agreement with one another, the three datasets allow robust provenance proposals to be put forward for consideration and debate.

## Analytical methods

Samples were taken from the 21 specimens selected for the research (Table 2), and the LI ratios were measured on a small aliquot of metal extracted from them (in the range 1.0–2.5 mg). The metal was dissolved in hot triply distilled concentrated nitric acid by high-pressure microwave digestion in sealed PTFE vessels. As described in Villa [91], the dissolved lead was purified using the SrSpec™ resin (EIChroM Industries). About 100ml of resin are commonly filled in a 3-mm diameter hand-made PTFE column. The height-to-width ratio is approximately 4. The sample solution is loaded in 0.5 ml 1M $HNO_3$, 1.5 ml of which is also used to wash out the matrix metals, while Pb is very strongly retained on the resin; Pb is then eluted with 3 ml 0.01M $HNO_3$ and is ready for analysis.

The measurement of LI ratios were performed using a Thermo Scientific Neptune multi-collector ICP-MS instrument at the Institut für Geologie, University of Bern (Switzerland). This instrument is equipped with a double-focusing geometry. The Faraday collector array allows the simultaneous acquisition of masses 202 to 209. The sample introduction system consisted of an auto-aspirating low-flow (50 ml/min) Apex de-solvating nebulizer (ESI Scientific, Omaha, NE, USA) mounted onto a combined cyclonic/double-pass spray chamber made of quartz glass. Potential isobaric interference of $^{204}Hg$ on $^{204}Pb$ was controlled and, if

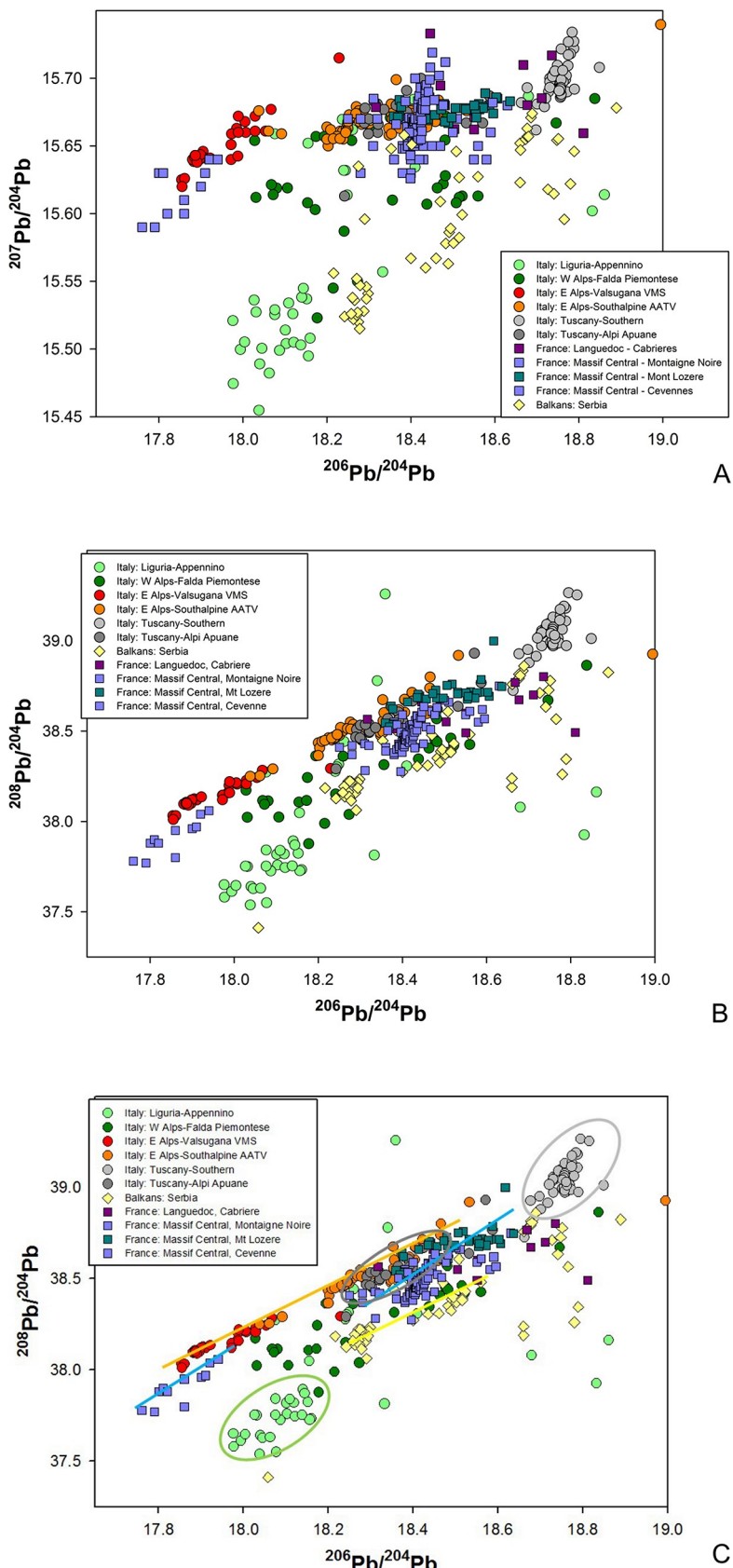

**Fig 6. LI diagrams comparing the copper ore fields of central Italy with other major European ore fields;** (a) 2D plot of $^{206}$Pb/$^{204}$Pb vs $^{207}$Pb/$^{204}$Pb LI data; (b) 2D plot of $^{206}$Pb/$^{204}$Pb vs $^{208}$Pb/$^{204}$Pb LI data; (c) 2D plot of $^{206}$Pb/$^{204}$Pb vs $^{208}$Pb/$^{204}$Pb LI data with marks showing ore fields (grey: Tuscany; pale green: Liguria-Apennine; dark grey: Apuanian Alps) and trend lines (orange: South-Eastern Alps; blue: Massif Central; yellow: Serbia).

necessary, corrected for by monitoring the $^{202}$Hg signal. Hydride formation (PbH) was monitored on mass 209 and never detected. Mass fractionation was monitored by adding a small quantity of Tl, which has a known $^{203}$Tl/$^{205}$Tl ratio, fractionated by the same mechanism as Pb and does not interfere with Pb isotope measurements [92]. The measurement accuracy was controlled with frequent measurements of the NIST SRM 981 standard reference material alternating with the sample measurements. The measured isotopic composition for SRM 981 was indistinguishable from the certified value and the recent, precise measurements available in the literature [93], so that no adjustment of the measured ratios was necessary. The external reproducibility on the SRM 981 reference material over the measuring period of several days amounted to ±0.015% (2s). The measured LI ratios and error estimates are reported in Table 4.

The experimental LI ratios measured on the central Italian metal objects discussed in this paper were compared with the extensive database of LI ratios of copper ores developed by the Department of Geosciences, University of Padua. The database encompasses the databases previously available in the literature, such as the OXALID database developed at the Oxford Isotrace Laboratory [94], complemented by the vast literature cited by Ling et al. [95], and the measurements performed by our research group.

The database has been enquired using different techniques. Beside the commonly employed 2D projections of the 3D isotopic space, the Euclidean Distances (thereafter EDs) [96] from each object to all copper ore data in the database were computed to estimate the more likely deposits supplying the metal, i.e. the ones having the shortest EDs. The deposits having a minimum of five ore samples at short distances from the object (within the 15 shortest computed EDs for each sample) were selected as the most likely sources and preliminary plotted as 2D diagrams (Fig 7). Each ore deposit, carefully defined from consistent geological, geochemical, and mineralogical parameters [21–22] was then used as a model for the calculation of the Kernel Density Estimations (thereafter KDEs) [97] and compared to the isotopic distribution of the object data (Fig 8). For objects lying in areas of strong overlap between ore fields, the most likely provenance was selected on the basis of: (a) KDE model density at the object point, and (2) compatibility of the object chemistry with the known geochemical and mineralogical character of the deposit [98]. The results thus obtained have further been assessed based on a number of archaeological considerations including the earliest evidence for metal production in the region being discussed, the local or non-local typology of the object, evidence of prehistoric ore-mineral extraction, and others, as discussed below.

## Results and discussion

### Objects with suggested provenance from central/southern Tuscany

The suggested provenances of the objects discussed in this paper are summarised in Table 5, along with their chemical signatures comprising major and minor elements as available from the literature. It is clear from the ED calculations and the 2D LI plots shown in Fig 7 that most objects (11 out of 20, or 55%) are isotopically compatible with the Southern Tuscan ore field as defined in Table 3. It is also observed that most objects in this group are composed of either (relatively) pure or arsenic-rich copper, while a single axe-head features low As and (relatively) high Sb, with notable Ag (Table 5:85). This is in line with the ore geology of the Southern

**Table 4. Lead isotope ratios of the objects analysed for the research.**

| DB ID | Lab ID | Site | Object | $^{206}Pb/^{204}Pb$ | ±2σ | $^{207}Pb/^{204}Pb$ | ±2σ | $^{206}Pb/^{204}Pb$ | ±2σ |
|---|---|---|---|---|---|---|---|---|---|
| 385 | Sel-Ax | Selvena | Axe | 18.5950 | 0.0033 | 15.6899 | 0.0027 | 38.8194 | 0.0077 |
| 360 | PSi7-Ax | Province of Siena | Axe | 17.7665 | 0.0032 | 15.5996 | 0.0032 | 37.7880 | 0.0064 |
| 100 | PGr-Ax | Province of Grosseto | Axe | 18.2631 | 0.0020 | 15.6491 | 0.0020 | 38.3588 | 0.0062 |
| 425 | Pal8-Ax | Sarteano, Palazzone | Axe | 18.7424 | 0.0048 | 15.7205 | 0.0039 | 39.0728 | 0.0103 |
| 126 | Te7-Ax | Il Teso | Axe | 18.7815 | 0.0026 | 15.7179 | 0.0027 | 39.1183 | 0.0079 |
| 127 | Te8-Ax | Il Teso | Axe | 18.6890 | 0.0060 | 15.7050 | 0.0060 | 38.9557 | 0.0126 |
| 355 | Fi-Ax0 | Province of Florence | Axe | 18.7720 | 0.0014 | 15.7204 | 0.0016 | 39.1181 | 0.0050 |
| 88 | Fi-Ax1 | Province of Florence | Axe | 18.7819 | 0.0020 | 15.7179 | 0.0019 | 39.1124 | 0.0054 |
| 85 | Fi-Ax9 | Province of Florence | Axe | 18.7279 | 0.0018 | 15.7013 | 0.0002 | 38.9606 | 0.0053 |
| 55 | Gar-Axg | Garfagnana | Axe | 18.0670 | 0.0018 | 15.5989 | 0.0020 | 38.0202 | 0.0061 |
| 357 | BM-00 | Near Terni | Axe | 18.7581 | 0.0029 | 15.7063 | 0.0027 | 39.0277 | 0.0081 |
| 358 | BM-01 | Near Terni | Axe | 18.7258 | 0.0033 | 15.7071 | 0.0024 | 39.0180 | 0.0060 |
| 412 | BM-26 | Abruzzo | Axe | 18.2526 | 0.0028 | 15.5548 | 0.0024 | 38.0405 | 0.0065 |
| 422 | BM-47 | Corneto (Tarquinia) | Axe | 18.7296 | 0.0016 | 15.6995 | 0.0016 | 38.9903 | 0.0049 |
| 170 | Que-Pg | Querceto | Dagger | 18.1964 | 0.0017 | 15.6444 | 0.0019 | 38.3193 | 0.0057 |
| 7 | Pz-Pg0 | Pianizzoli | Dagger | 18.7814 | 0.0011 | 15.7198 | 0.0016 | 39.1374 | 0.0036 |
| 8 | Pz-Pg2 | Pianizzoli | Dagger | 18.3396 | 0.0033 | 15.6628 | 0.0029 | 38.4891 | 0.0077 |
| 71 | Vet-Al | Vetralla | Halberd? | 18.6221 | 0.0025 | 15.6723 | 0.0022 | 38.8102 | 0.0057 |
| 39 | Pz-Al8 | Pianizzoli | Halberd | 18.4429 | 0.0043 | 15.6511 | 0.0040 | 38.5058 | 0.0103 |
| 39A | Pz-R3 | Pianizzoli | Rivet of Pz-Al8 | 18.8111 | 0.0019 | 15.7179 | 0.0018 | 39.1113 | 0.0053 |
| 38 | Pz-Al9 | Pianizzoli | Halberd | 18.7432 | 0.0020 | 15.6989 | 0.0022 | 39.0239 | 0.0075 |

Pb isotopic ratios and error estimates of the metal objects investigated for the research. Numbers in the DB ID column (first left) refer to the database of early central Italian metals compiled by Dolfini [50,66], while the Lab ID column (second from left) lists the sample lab codes used by the University of Padua.

Tuscan deposits, which are rich in chalcopyrite, bornite, chalcocite, and tennantite, along with other copper-bearing minerals [76,82,87]. Furthermore, the pure-Cu and Cu-As composition of these objects is consistent with the chemistry of the analysed metal droplets from San Carlo-Cava Solvay, the 4th millennium BC smelting site discussed above [19]. Taken together, these data contribute a strong case towards the local origin of the copper metal, bearing in mind that, in this context, 'local' means several ore sources from a fairly wide area of middle and lower Tuscany (including the Elba Island), which can hardly be discriminated further based on LI alone.

The objects of likely Tuscan origin encompass nine axe-heads, one dagger, and a halberd (plus the halberd rivet analysed). Looking at their find spots provides an opportunity to assess, and nuance, notions of local production and circulation. Pianizzoli, for example, lies within the Massetano district, a short distance away from major ore bodies bearing similar LI fingerprints as the dagger and halberd from the site (Table 5:7 and 38). Interestingly, both the dagger (type Massa Marittima) and the halberd (type Montemerano) belong to typological groupings overwhelmingly concentrated in west-central Italy [3:plate 102]. Regardless of the exact origin of the copper metal, it is hard to dispute that they would have been made and exchanged within the region.

Other objects, however, were found further afield from their likely ore sources. This is the case with six axe-heads from Sarteano (Siena), Il Teso (Arezzo), and the Florence area (Table 5:85,88,126,127,355 and 425); their closest isotopically compatible ore bodies lie in the Colline Metallifere, 50–70 km (as the crow flies) to the north-west and south-west, respectively. Distances increase further with three axe-heads from the hinterland of Terni (a town in inner

**Table 5. Chemical composition and suggested provenance of the objects analysed for this project.**

| DB ID | Lab ID | Site | Object | Chemical analysis | Method | Cu | Sn | Pb | As | Sb | Ag | Ni | Bi | Au | Zn | Co | Fe | S | Suggested provenance |
|---|---|---|---|---|---|---|---|---|---|---|---|---|---|---|---|---|---|---|---|
| 7 | Pz-Pg0 | Pianizzoli | dagger | Junghans et al. 1960, 609 | OES | - | tr. | 0.14 | 2.60 | 0.03 | 0.02 | <0.01 | 0.01 | 0 | 0.02 | 0 | 0.02 | - | Southern Tuscany |
| 8 | Pz-Pg2 | Pianizzoli | dagger | Junghans et al. 1960, 610 | OES | - | 0 | 0 | 3.50 | 0 | 0.05 | 0 | 0.03 | 0 | 0 | 0 | 0 | - | Tuscany / Massif Central (France) |
| 38 | Pz-Al9 | Pianizzoli | halberd | Junghans et al. 1960, 611 | OES | - | 0 | 0.02 | 2.20 | tr | 0.10 | tr | 0.11 | 0 | 0 | 0 | tr | - | Southern Tuscany |
| 39 | Pz-Al8 | Pianizzoli | halberd | Junghans et al. 1960, 612 | OES | - | 0 | 0.04 | 2.75 | 0.45 | 0.10 | 0.02 | 0.001 | 0 | 0 | 0 | 0 | - | Tuscany / Massif Central (France) |
| 39A | Pz-R3 | Pianizzoli | rivet of halberd no.39 | Artioli & Angelini, unpublished | EPMA | yes | | | no | no | | | | | | | | | Southern Tuscany |
| 55 | Gar-Axg | Garfagnana | axe | Junghans et al. 1960, 604 | OES | - | 0 | 0 | 0.36 | 0.22 | 0.55 | tr | 0.01 | 0 | 0 | 0 | tr | - | Falda Piemontese (Western Alps) |
| 71 | Vet-Al | Vetralla | halberd? | Junghans et al. 1974, 20599 | OES | - | 0 | 0 | tr | 0.06 | 0.27 | 0 | 0 | 0 | 0 | 0 | 0 | - | Apuanian Alps (NW Tuscany) |
| 85 | Fi-Ax9 | Province of Florence | axe | Junghans et al. 1960, 597 | OES | - | 0 | 0 | 0.03 | 0.85 | 0.52 | tr | 0 | 0 | 0 | 0 | 0.05 | - | Southern Tuscany |
| 88 | Fi-Ax1 | Province of Florence | axe | Junghans et al. 1960, 594 | OES | - | 0 | 0 | 0 | 0.25 | 0.06 | tr | 0 | 0 | 0 | 0 | 0.09 | - | Southern Tuscany |
| 100 | PGr-Ax | Province of Grosseto | axe | Junghans et al. 1960, 606 | OES | - | 0 | 0 | 1.7 | 2.7 | 0.44 | 0.04 | ~0.4 | 0 | tr | 0 | 0 | - | Falda Piemontese (Western Alps) |
| 126 | Te7-Ax | Il Teso | axe | Otto & Witter 1952, 32 | AAS | 99.9 | 0 | tr | 0 | tr | tr | tr | 0 | 0 | 0 | 0 | tr | - | Southern Tuscany |
| 127 | Te8-Ax | Il Teso | axe | Otto & Witter 1952, 33 | AAS | 99.9 | 0 | 0 | 0 | tr | tr | tr | 0 | 0 | 0 | 0 | 0 | - | Southern Tuscany |
| 170 | Que-Pg | Querceto | dagger | Junghans et al. 1960, 636 | OES | - | <0.01 | 0.02 | 2.1 | 0.9 | 0.13 | 0.02 | tr | 0 | 0 | 0 | 0.02 | - | Falda Piemontese (Western Alps) |
| 355 | Fi-Ax0 | Province of Florence | axe | Junghans et al. 1960, 596 | OES | - | 0 | 0 | 0 | 0 | 0 | 0 | 0 | 0 | 0 | 0 | 0.06 | - | Southern Tuscany |
| 357 | BM-00 | Near Terni | axe | Hook 1999, 2007 | ICP-AES | 100.0 | <0.01 | 0.01 | <0.01 | <0.01 | 0.047 | 0.007 | 0.058 | - | <0.01 | <0.003 | <0.003 | <0.01 | Southern Tuscany |
| 358 | BM-01 | Near Terni | axe | Hook 1999, 2007 | ICP-AES | 100.0 | <0.01 | <0.01 | 0.02 | 0.07 | 0.035 | 0.006 | <0.01 | - | <0.02 | <0.003 | <0.003 | <0.01 | Southern Tuscany |
| 360 | PSi7-Ax | Province of Siena | axe | Junghans et al. 1960, 654 | OES | - | 0 | 0 | tr | 0.07 | 0 | 0 | tr | 0 | 0 | 0 | 0 | - | Massif Central (France) |
| 385 | Sel-Ax | Selvena | axe | Artioli & Angelini, unpublished | EPMA | yes | | | no | no | | | | | | | | | Apuanian Alps (NW Tuscany) |
| 412 | BM-26 | Abruzzo | axe | Hook 2007 | ICP-AES | 100.9 | <0.01 | <0.01 | <0.01 | <0.01 | 0.022 | <0.003 | <0.01 | - | <0.01 | <0.003 | 0.005 | <0.01 | Falda Piemontese (Western Alps) / Serbia |
| 422 | BM-47 | Corneto (Tarquinia) | axe | Hook 2007 | ICP-AES | 94.5 | 0.01 | 0.03 | 1.81 | 0.02 | 0.088 | 0.01 | 0.12 | - | <0.01 | <0.002 | <0.005 | <0.01 | Southern Tuscany |
| 425 | Pal8-Ax | Sarteano, Palazzone | axe | Artioli & Angelini, unpublished | EPMA | yes | | | no | no | | | | | | | | | Southern Tuscany |

Chemical composition and suggested provenance of the objects analysed for the research. Find spots, chronology, and archaeological data are listed in Table 3. Objects ID 385, 425 and 39A were first analysed for their alloy composition by our team; we provide here preliminary presence/absence data for Cu, As and Sb.

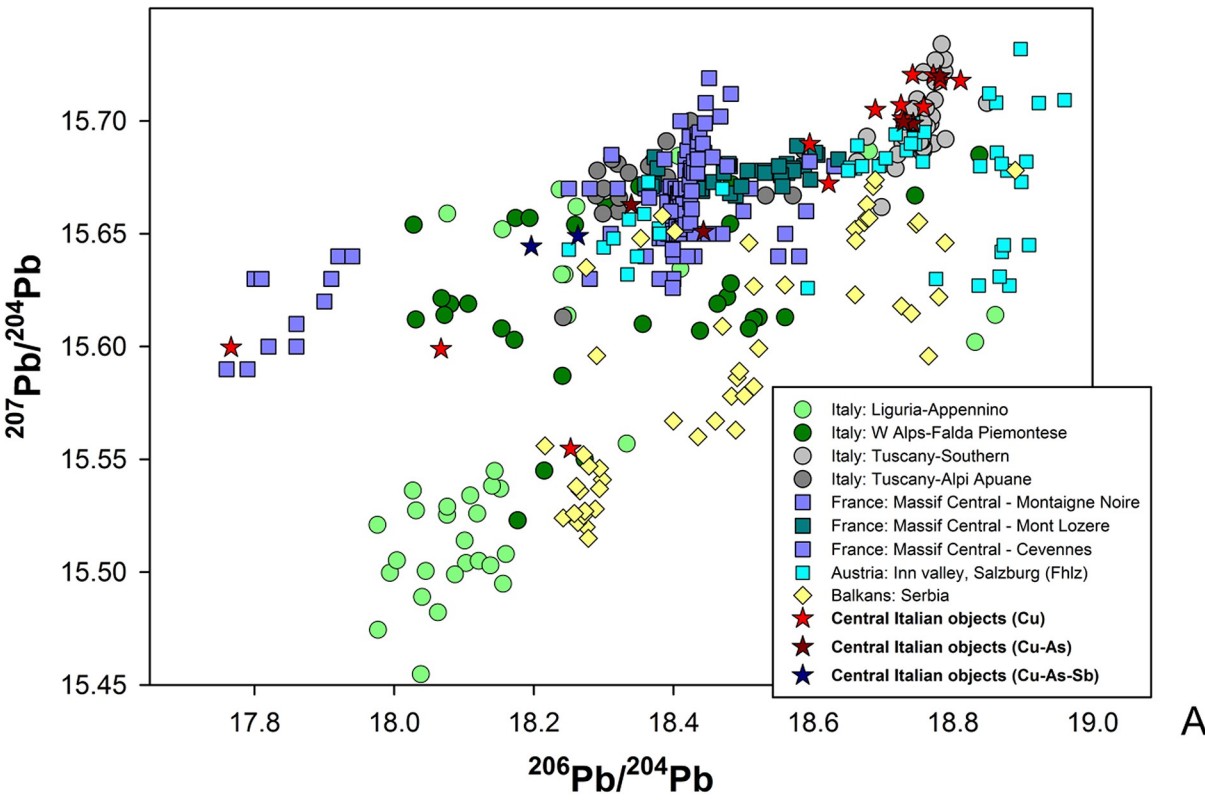

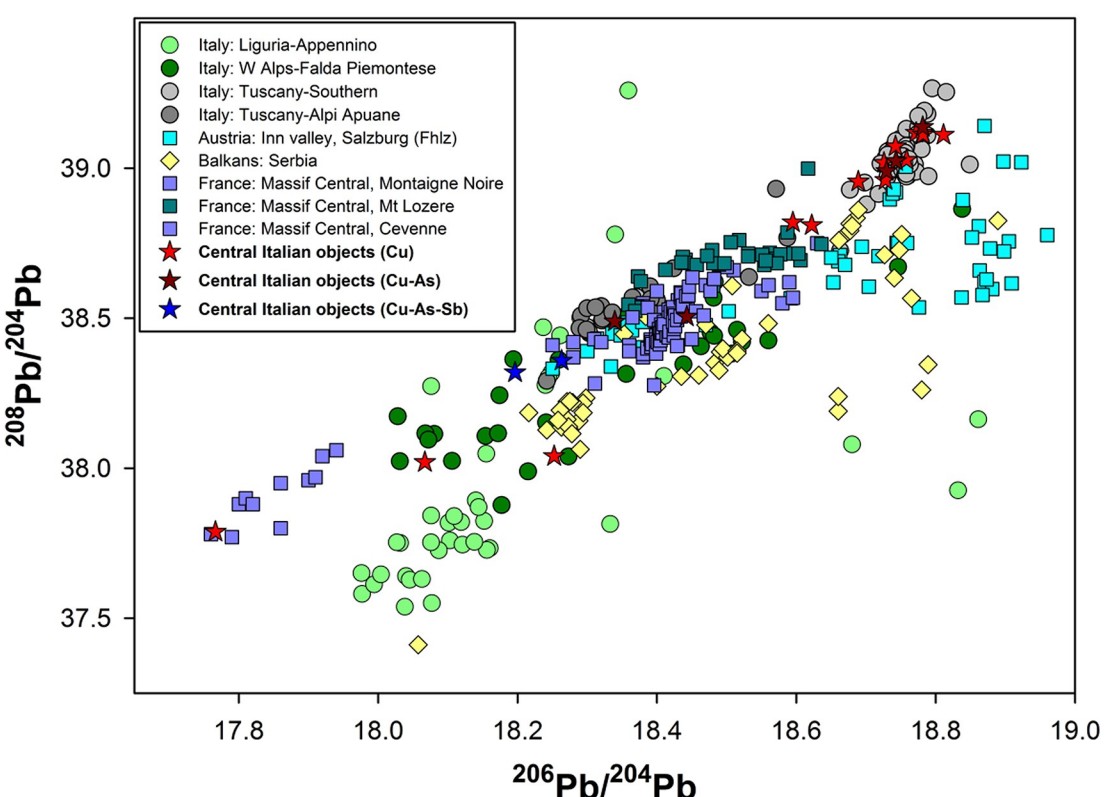

**Fig 7. LI signatures of central Italian metal objects;** (a): 2D plot of $^{206}Pb/^{204}Pb$ vs $^{207}Pb/^{204}Pb$ LI data; (b) 2D plot of $^{206}Pb/^{204}Pb$ vs $^{208}Pb/^{204}Pb$ LI data.

Umbria) and Corneto (modern-day Tarquinia; Table 5:357,358 and 422). For them, the closest possible ore sources lie some 120–150 km away to the north and north-west. To make sense of these implements we must postulate the existence of exchange networks of regional scope. It is of course possible that the distance between mines and deposition locales was covered cumulatively over extended periods of time, and may have involved multiple agents acquiring, using, and passing the objects down the line. These may not even be the objects originally cast from the smelted ore. Indeed, any recycling of metal that did not involve the addition of alien copper would have preserved the original LI signature of the mine. Certainly, none of the axe-heads from this subgroup show typological traits suggesting origins from outside Italy.

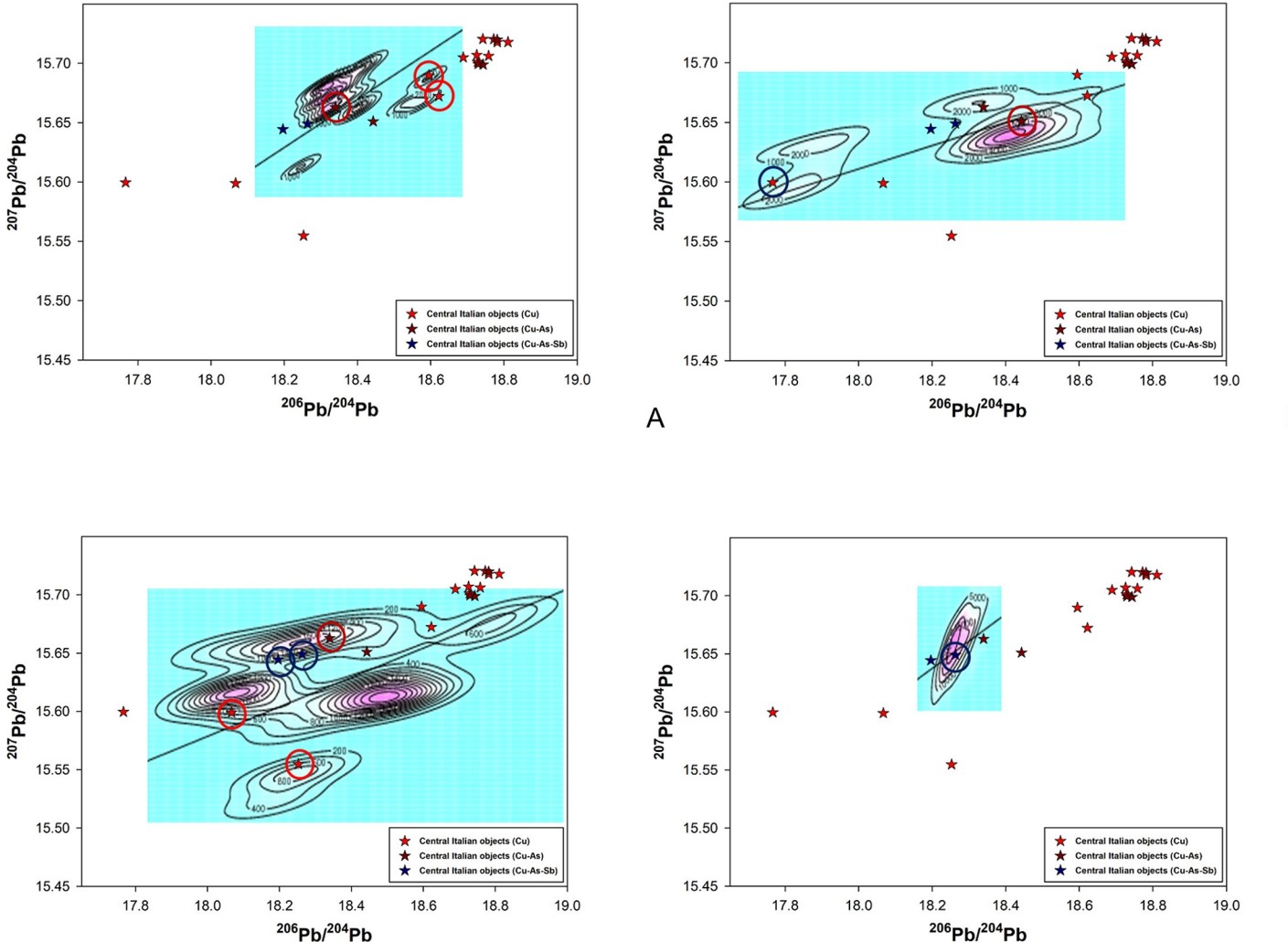

**Fig 8. LI diagrams comparing central Italian metal objects with the KDE models of the most likely source ore deposits based on Euclidean Distances.** (a) Apuanian Alps model; (b) French Massif Central model; (c) Western Alps–Falda Piemontese model; (d) Sardinia (centre-Sulcis area) model.

## Objects with suggested provenance from northwest Tuscany and objects of uncertain Tuscan origin

The axe-head from Selvena, southern Tuscany (Table 5:385), and the unique halberd from Vetralla, northern Latium (Table 5:71), display LI signatures lying at the boundary of the KDE model of the Southern Tuscan ore domain. They also have a low statistical fit with the KDE models of the Western Alps and the French Massif Central. The best fit is provided by the KDE model for the Apuanian Alps, northwest Tuscany (Fig 8A). Tentative provenance from this area is therefore proposed on statistical grounds. Unfortunately, chemistry is of little aid in building a robust provenance hypothesis as both objects are made of rather pure copper, with low Ag for the halberd. They are both assigned to the mid-4th millennium BC on morphological and technological grounds, bearing in mind that early axe-heads might be difficult to date accurately, while the Vetralla halberd is a unique artefact lacking contextual data and *comparanda*.

Assuming that the provenance hypothesis cautiously proposed here is correct, these objects would testify to the sourcing of Apuanian copper from the early Copper Age. There is currently no evidence of prehistoric ore mining in this area, but absence of evidence should not be taken as evidence of absence, for early workings were all but wiped out by intensive historic mining (including modern open casting) all over Tuscany [10,99]. Provenance from the Apuanian Alps would denote prehistoric exchange networks stretching 150–250 km away from the copper sources. This is certainly possible both in itself and in light of the data discussed below. However, considering the uncertainties surrounding the KDE model, we should perhaps treat this as a working hypothesis pending further LI analyses involving a larger sample size.

Two more objects, a dagger and a halberd from Pianizzoli, southern Tuscany, bear LI signals compatible with the Apuanian copper deposits (Table 5:8 and 39; Fig 8A). However, their signals fall in a region of strong overlap between isotopic ore fields, intersecting the French Massif Central, the Western Alps, the Swiss Valais, and Southern Tuscany (Fig 8B). Under the circumstances, no credible provenance hypothesis can be proposed on LI data alone. Archaeology, however, sheds a modicum of light on the problem. The dagger belongs to the 'Massa Marittima' typological family, dated to the middle Copper Age (3350–2800 BC). Chemically, it features Cu-As alloy composition that is typical of bladed implements from the region (Table 5:8). The halberd is classified under the 'Montemerano' type and dated to the mid/late Copper Age (3350–2200 BC); it has similar Cu-As alloy composition as the dagger, with 0.45wt% Sb (Table 5:39).

Interestingly, both objects hail from a stratified burial deposit that has yielded two more objects (a dagger and a halberd) seemingly cast from local copper (Table 5:7 and 38; see above for discussion). The same is true of the halberd rivet analysed–something that, in and of itself, implies local manufacture (Table 5:39A). How to make sense of isotopic differences between otherwise very similar objects from the same site? An economical explanation is that they were all made from Southern Tuscan copper, some of which, however, came from a source whose LI signature is more shifted towards lower Pb values. However, the atypical LI signatures of these objects could also result from the mixing of copper from different sources. In particular, the mixing of southern Tuscan copper with metal from a more westerly source, be it the Apuanian Alps, the central/western Alps, or the French Massif Central, would likely produce the signal we see here. Overall, it is not implausible that mid/late Copper Age objects would be cast from mixed metal considering that copper production intensified in the early Copper Age, and Western Alpine mines might have been worked from the same time (see below). Enough metal would have been in circulation by this time to warrant early copper mixing. All

## Objects with suggested provenance from outside Tuscany

All other objects in the sample (five out of 20, or 25%) display LI ratios that are not compatible with the ore fields of Southern Tuscany or the Apuanian Alps. Among them are a trapezoidal axe-head from the Grosseto area and a Guardistallo-type dagger from Querceto, Siena (Table 5:100 and 170). Both objects have high-As, high-Sb alloy compositions that are otherwise typical of Copper Age metalwork from central Italy. Indeed, this is the most distinctive chemical signature of 'Rinaldone metals', since ternary Cu-As-Sb alloys are not documented in the northern or southern peninsula, or Sardinia [9–10,100–101]. It is thus surprising to learn that the copper used for these objects may not be from Tuscany, considering the abundance of polymetallic sulphides in the region (see above).

Where might the copper metal hail from? The LI signal of the Grosseto axe fits very well with the KDE model of Sardinian ores (Fig 8D), but we exclude this provenance due to the object's high-Sb content. Antimony is generally absent in isotopically compatible copper-bearing deposits from the region, including Genna Olidoni and Funtana Raminosa. None of these sources have yielded Sb-containing minerals, except for very minor traces of berthierite [102–103]. This reading is strengthened by the analysis of Bronze and Iron Age Sardinian metals, featuring similar LI signals but tiny amount of Sb, typically less than few tens of ppm [104]. Our As/Sb-rich objects are also isotopically compatible with the KDE model of the Western Alps (i.e. Falda Piemontese and Brianzonese Ligure ore-mineral deposits; Fig 8C, blue circles) and with Iberian ore bodies from the Alcudia Valley and Iberian Pyrite Belt (not shown in the graphs). Considering that As and/or Sb impurities are present in several cupriferous ore bodies from the Western Alps (e.g. in the tetrahedrite-tennantite compounds from the Murialdo-Pastori mine near Savona, western Liguria [90,105]) provenance from this region is preferred to Iberia on grounds of proximity. The Western Alpine isotopic field comprises a broad arc ranging from central/western Switzerland to western Liguria, which can hardly be discriminated further based on LI fingerprinting alone.

Typologically, the Grosseto axe-head is indistinctive. The Guardistallo-type dagger from Querceto, however, is idiosyncratically central Italian [3]; we must thus presume that it was cast in Tuscany using metal from the Western Alps. While the axe-head is hard to date precisely due to its unspecific morphology, the dagger is firmly tied to the early/middle Copper Age (3600–2800 BC) by several radiocarbon dates and tight *comparanda* [11,71]. As the situation stands, copper mining is documented in the Western Alps from the late 3rd millennium BC (e.g. at Saint-Véran) [27,106], although the first metal artefacts appeared in the region a thousand years earlier [107–108]. This is compatible with the dagger's chronology proposed here, which, if accepted, would independently pinpoint the beginning of copper sourcing in the Western Alps to the 4th (or initial 3rd) millennium BC.

Two further objects show LI signatures statistically compatible with Western Alpine ore deposits (Fig 8C, red circles). The first is an axe-head from Garfagnana, a mountainous region in northwest Tuscany (Table 5:55). This is a morphologically unique implement for which no tight comparisons can be drawn with any Italian axe-head known to date. Its unusual shape might suggest importation from outside the peninsula, although unique objects do occur, in small numbers, in the early Italian record (e.g. [109] and the Vetralla halberd discussed above). The object's archaic design and features might indicate 4th millennium production. Chemically, it displays medium values of As, Sb and Ag; this is compatible with sourcing from Western Alpine polymetallic ores (or the co-smelting of Cu and As/Sb ores from this area). Isotopic

and chemical signals seem therefore to support each other and strengthen the provenance hypothesis proposed here.

The second object is a trapezoidal axe-head from Abruzzo, a region in east-central Italy (Table 5:412). The object falls within a broad class of stout tools with bi-convex profiles that are overall dated to the early Copper Age, 3600–3350 BC [71]. Isotopically, it is compatible with the Western Alpine ore deposits; its LI signal, however, partly overlaps with several Balkan ore fields, and is especially close to the Timok magmatic area in Serbia (Fig 7). Chemically, the implement is made of very pure copper. This is consistent with both chalcopyrite and bornite-dominated ores from the Western Alps (e.g. Saint-Véran, Beth-Ghinivert, and Via Fiorcia) [90] and the malachite/azurite secondary ores of the Serbian mines (e.g. Bor and Majdanpek) [110]. Although alloy composition and chemistry are of little help in resolving the provenance ambiguity, it is intriguing that the sole object in the sample showing possible Balkan connections is from east-central Italy. This invites further research into early metals from the region.

Our sample also comprises an axe-head from the hinterland of Siena showing very low $^{206}Pb/^{204}Pb$ and $^{208}Pb/^{204}Pb$ values (Table 5:360). Such isotopic values are compatible with very few ore deposits sampled to date. Based on the ED calculations, only two areas can be suggested as sources of the copper metal: the Alcudia Valley in the Iberian Peninsula and the Montaigne Noire district of the Massif Central, southern France. Based on the KDE estimates of the available samples, the French provenance is statistically preferred (Fig 8B). The implement is typologically dated to the early Copper Age (3600–3350 BC), although a lower chronology cannot be ruled out completely. Chemically, it features a rather pure alloy composition showing detectable, if low, Sb values; this hints at an ore source containing this element as an impurity. Antimony is frequently found in the copper-bearing deposits of the Massif Central, and shows in early metals from the region [107,111]. Provenance from southern France is thus a distinct possibility.

Learning that the copper used to cast an early axe-head from Tuscany might come from the French Midi is an unexpected finding provoking further reflections. The first is chronological. Whilst we acknowledge that the object is not as securely dated as one would wish it were, it is most probably 4th millennium BC. As with the Western Alpine sources discussed above, this axe-head raises the possibility that the Massif Central ore deposits might have first been exploited somewhat earlier than 3100 BC [16,112]. The second is typological. Although early Italian axe-heads could be difficult to classify, the object falls within the morphological parameters of central Italian implements. This suggests that it was cast in Tuscany using copper metal that might have come from faraway lands.

Notably, several markers of a new 'Italian connection' appeared in the Western Alps and the lower Rhone valley from the mid/late 4th millennium BC. These include rock carvings (and an actual specimen) of Remedello daggers–a characteristically Italian dagger type [113]. The 'Italian connection' also encompasses a metal-rich burial from Fontaine-le-Puits, Savoy, which has yielded a metal dagger whose shape and alloy composition suggest provenance from central Italy. The burial itself is culturally alien in the area and shows striking similarities with 'Rinaldone-style' Italian interments. Two radiocarbon dates pin the burial to the 4th millennium BC, the highest probability falling between 3500–3340 BC [108]. In light of this evidence, it is not far-fetched to suggest that Italian prospectors and metalworkers active the Western Alps from the mid-4th millennium BC might have travelled on to the Massif Central, perhaps bringing with them knowledge of extractive metallurgy [12,114].

## Understanding metal procurement and exchange in early Italy

Early Italian metallurgy was long believed to result from socio-technological dynamics of largely regional scope. After its invention in the late 5th or early 4th millennium BC, perhaps triggered by knowledge transfer from the Balkans via the eastern Alps, extractive metallurgy–it was posited–would have taken hold in the ore-rich districts of Tuscany and surrounding regions from the mid-4th millennium BC [12]. Here, it would have grown and developed through the ages, nurtured by near-inexhaustible sources of copper and other sought-after metals. This prevalent view was supported by three arguments. First, early metals tend to cluster around the mining areas of west-central Italy; their numbers decline significantly as one moves into ore-starved lands east of the Apennines and south of the Tiber valley [115,50]. Secondly, early metals were overwhelmingly cast and fashioned according to regional design choices. This is especially the case with daggers and halberds, whose distinctive morphologies are taken to indicate local production and circulation, but is also true of certain axe-head types [3–5,71,116–117]. Thirdly, early metals, and especially daggers and halberds, often contain relatively high amounts of arsenic and/or antimony; this is normally explained with derivation from polymetallic compounds, of which Tuscany is rich. Notably, while binary Cu-As alloys are found all over the peninsula, ternary Cu-As-Sb alloys are near-unique to the 'Rinaldone' metallurgical style of central Italy. This would confirm the local character of early metal production and exchange in the region [9–10,100,115].

The data discussed in this paper have enriched, and nuanced, the orthodox view outlined above. Based on the provenance hypotheses proposed above, 55–75% of metals in our sample would be cast from Tuscan copper, from either Southern Tuscany or the Apuanian Alps. Overall, this is in line with the morphology and alloy composition of these objects, suggesting local procurement, manufacture, and uses. At least 25% of the objects, however, would be made from copper coming from further afield. This would have reached central Italy through exchange mechanisms linking the middle and upper Tyrrhenian coast, the western Alps, and southern France. The copper metal was probably exchanged as finished objects, considering that ingots are not documented in Italy until the Bronze Age [50,116,118]. In central Italy, the foreign-looking artefacts would be recast into 'Rinaldone-style' axe-heads, daggers, and halberds to satisfy cultural expectations concerning object design and function. In the 4th millennium BC, this would not normally involve any mixing of copper from different sources. Indeed, it is likely that early metal recycling hinged upon the recasting of individual objects into new objects of the same kind; axe-heads would be cast into new axe-heads, and so would daggers and halberds. This reading is not only suggested by the lack of blurred or anomalous LI signatures in the earliest metals from our sample, but also by their stable chemistry by object category. Had early smiths cast daggers into axe-heads and vice versa, we would not witness the sharp compositional patterns found in central Italian metals, with axe-heads being mostly pure-Cu and daggers and halberds being either Cu-As or Cu-As-Sb [9–10,100,115].

Seemingly, things started to change in the 3rd millennium BC. A dagger and a halberd in our sample display isotopic signatures that might be explained by admixture of copper from Southern Tuscany and a more westerly ore field (Table 5:8 and 39). Still, their high-As alloy composition suggests that recycling would be carried out preferentially within object category. This behaviour may have been motivated by distinctive cultural notions concerning early metallic substances, whereby pure copper, arsenical (and arsenical/antimonial) copper, and silver/antimony were conceptualised as distinct metals to be used for casting different objects which, in turn, would serve different purposes and acquire unlike social biographies and meanings [10,50,66]. Interestingly, this reading is in line with copper recycling practices in early Europe as revealed by the study of trace element behaviour in compositional groupings,

or 'Oxford system' [24,119–121]. In the British Isles, for example, metals were rarely recycled in the Chalcolithic (*c*.2500-2200 BC). When this occurred, they were normally cast into similar objects, ostensibly without adding any exogenous metal in the crucible. It was only in the Early Bronze Age that cultural taboos against breaking and melting old objects were challenged and recycling became both more frequent and less rule-bound. Growing amounts of metals with mixed chemical signatures and depleted volatile elements indicate this clearly [119,122]. This picture ties in neatly with the data discussed in this paper, showing, incidentally, that LI analysis and the 'Oxford system' complement one another rather well [24,123].

The interpretation proposed here is further strengthened by comparing the LI signals of central Italian metals with those of other 4th and 3rd millennia BC objects from the literature (Fig 9). These include, for Italy, four objects from the Col del Buson hoard, northeast Italy [124]; two awls from Arene Candide and Grotta della Pollera, Liguria [30]; and the Iceman's axe-head from the eastern Alps [125]; for Switzerland, metalwork from Saint-Blaise/Bains des Dames near Neuchâtel [126] and Zug-Riedmatt [127]; for Austria, several objects pertaining to the Münchshöfen cultural horizon as identified by Höppner and co-workers, including the Hornstaad disk [110]; for Serbia, metalwork assigned to the Baden-Kostolac horizon [128, groups 4 and 7], which is roughly contemporary with the Mondsee group from the northern Alps [129]; and for central Europe, a number of objects belonging to the Mondsee and Riesebusch horizons [130].

Based on the diagrams presented at Fig 9 we note that: (1) all objects from north-eastern Italy show LI signatures compatible with sources from the south-eastern Alps, except for an item from Col del Buson, possibly made from Balkan copper, and the Iceman's axe-head, whose LI signal is compatible with the ore fields of Southern Tuscany; (2) all objects from west-central and north-western Italy display LI signatures compatible with Tuscan (i.e. Southern Tuscany and Apuanian Alps), Western Alpine, and, in one instance, southern French ore deposits; (3) most Swiss objects can tentatively be provenanced from three major copper sources: the Alps (mostly the western Alps but also the Valsugana VMS ores in the south-eastern Alps), the Massif Central, and the Austrian falhores from the Inn Valley. However, the Zug-Riedmatt axe-head might be made from Tuscan copper, and some of the Blaise/Bains des Dames objects show LI signals compatible with Irish and Serbian mines [partly *contra* 126]; (4) apart from the Hornstaad disk (Lake Constance), virtually all objects from the central and eastern European Münchshöfen, Baden-Kostolac, and Mondsee horizons seem to be made of copper from either central Europe (Bohemian or Slovakian ores) or the Balkans.

These data suggest three distinct copper procurement and circulation networks: (1) central and eastern European communities would source their copper from the north-eastern Alps, the Slovakian Ore Mountains, and the central Balkans; (2) communities located in west-central and north-western Italy would obtain their copper from Tuscany, the Western Alps, and the French Midi; and (3) north-eastern Italian communities would tap into more self-contained circuits originating from the south-eastern Alps. The three networks seem to have operated largely independently from one another. Some metals did move across boundaries (e.g. the Iceman's axe-head) [125], but these are few and far between. As the situation stands, the sole mould-breakers were late Neolithic societies from the central Alps. Perhaps due to their location at the crossroads of the three networks, they could tap into them all, reaching out, on occasion, as far as the French Midi to the southwest, Tuscany to the south, and perhaps Serbia to the east. Chronological factors, as well as geography, might contribute to explaining the unique central Alpine pattern (especially in light of the hiatus in metal production affecting the region in the late 4th millennium BC) [131], but their investigation goes beyond the scope of this paper. What we remark here is that the three networks mostly operated within their own bounds, meeting and mingling only in the central Alps.

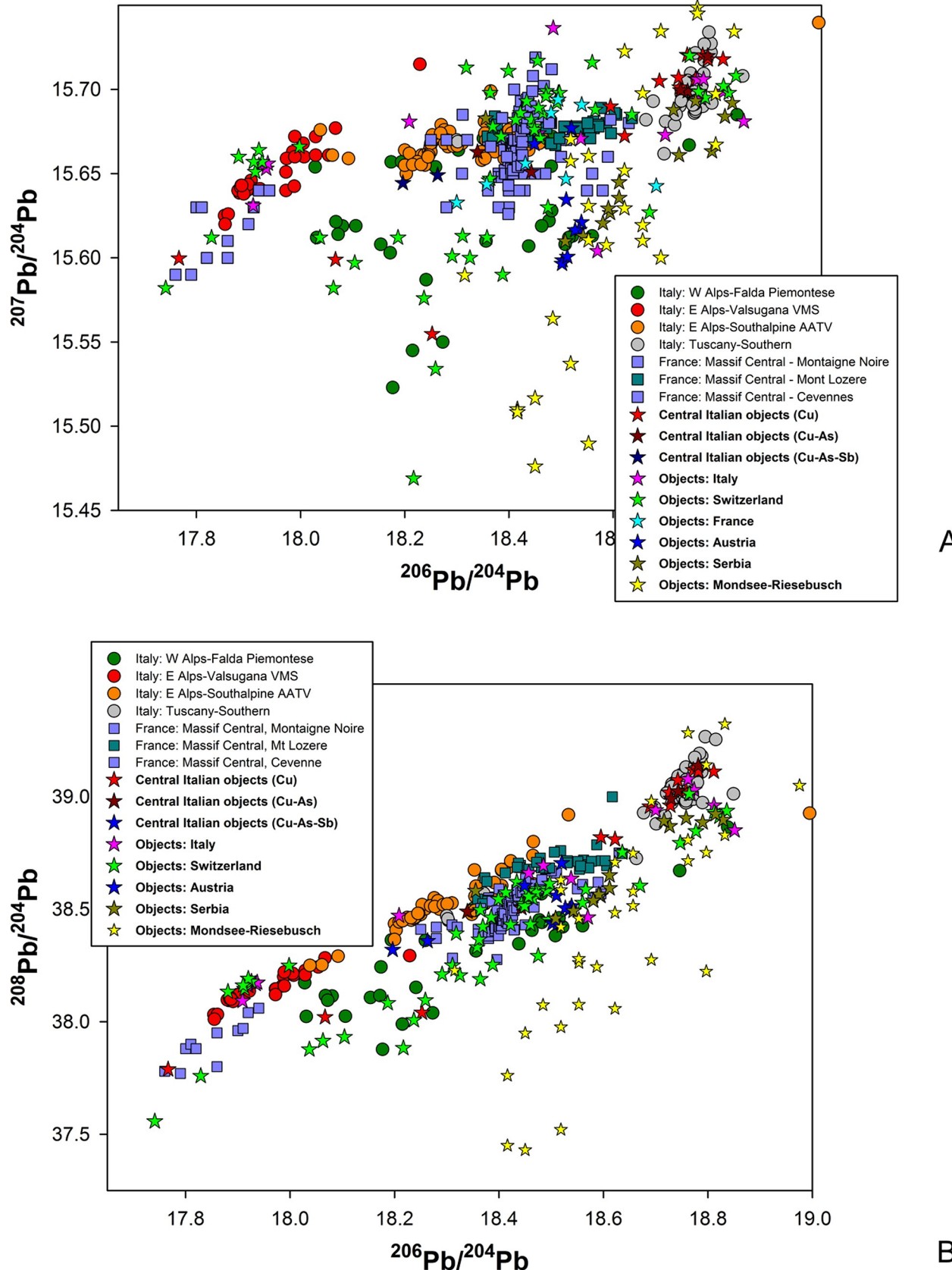

**Fig 9. Comparison of the LI data of 4ʰ and 3ʳᵈ millennia BC objects from Italy, Switzerland, Austria and Serbia, with ore data**; (a): 2D plot of ²⁰⁶Pb/²⁰⁴Pb vs ²⁰⁷Pb/²⁰⁴Pb LI data; (b) 2D plot of ²⁰⁶Pb/²⁰⁴Pb vs ²⁰⁸Pb/²⁰⁴Pb LI data.

Interestingly, the 'Oxford system' reconstruction of Alpine metal flow corroborates the picture painted by lead isotopy [23,132]. By examining variation in metal chemistry over time, Perucchetti and collaborators (including this paper's lead author) concluded that, in the Alpine region: (a) most metals were procured, exchanged, and consumed locally during the late Neolithic and Copper Age; exceptions to the rule are not quantified, but they might not be far from the 75–25 ratio of local v. alien metal proposed in this paper; (b) variation in alloy composition decreases in the Early Bronze Age, pointing to growing exchange and admixture within broader (and largely mutually exclusive) circuits in the western and eastern Alps; (c) tin-bronze metallurgy first emerged in the Early Bronze Age, with tin moving independently from copper but following, to a great extent, the east-west split highlighted above. Remarkably, the split does not follow the Alpine watershed, but cuts it asunder from the Swiss plateau to the Po valley.

The similarities between Perucchetti and collaborators' reading and the model proposed here are striking, considering that they were developed using dissimilar conceptual frameworks and analytical methods. However, this study integrates and nuances their findings in three important ways. Firstly, it backdates the east-west split in copper circulation zones to the Copper Age, bringing forwards one of the most stable and enduring cultural boundaries of early Europe, which cuts across physical geography and, in the Po valley, a navigable river system. Secondly, it extends this cultural boundary southwards into central Italy, suggesting that communities living west of the Apennines would partake in the exchange network linking Tyrrhenian Italy with the western Po plain, the Western Alps, and southern France. It is presently unclear if the communities living east of the Apennines would tap into this circuit. It is fascinating to note, however, that the one object analysed from this area shows a LI fingerprint compatible with both Western Alpine and Serbian ore deposits (Table 5:412); this might point to a cross-Adriatic circulation sphere worth investigating further. Both sides of the Adriatic were drawn into the sprawling 'Cetina' cultural network in the mid/late 3ʳᵈ millennium BC, and it is tempting to postulate that the seeds of cross-Adriatic connectivity were sown a thousand years earlier [133–134]. Thirdly, our research shows with greater clarity than in Perucchetti et al. [23] the unique advantage enjoyed by central Alpine communities, which would have tapped into all three circuits from the beginning of the metal age.

As a final point to make, our research has provided important evidence concerning the Italian ore-mineral sources that early metalworkers *did not* exploit to make copper. The most surprising result is that none of the metal objects analysed bears the isotopic signal of the ophiolitic mines of eastern Liguria (e.g. Libiola and Monte Loreto), which were first worked in the mid-4ʰ millennium BC (Fig 10) [1–2,135]. Despite our wide-ranging sampling extending to nearby ore deposits, and the statistically significant model developed as part of our project, this copper remains elusive. Likewise, we have not identified any copper from the ore deposits of the Tusco-Emilian Apennines (e.g. Impruneta), the Tuscanian Units of the inner Ligurides (e.g. Montecatini-Val di Cecina), and the Umbro-Marches Units of the outer Ligurides (e.g. Monti Rognosi) (Table 3). Considering the small sample size, however, future research might be able to capture some of this metal.

## Conclusions

The article has discussed results of an interdisciplinary research project investigating the provenance and exchange of 4ʰ and 3ʳᵈ millennium BC metal objects from central Italy through

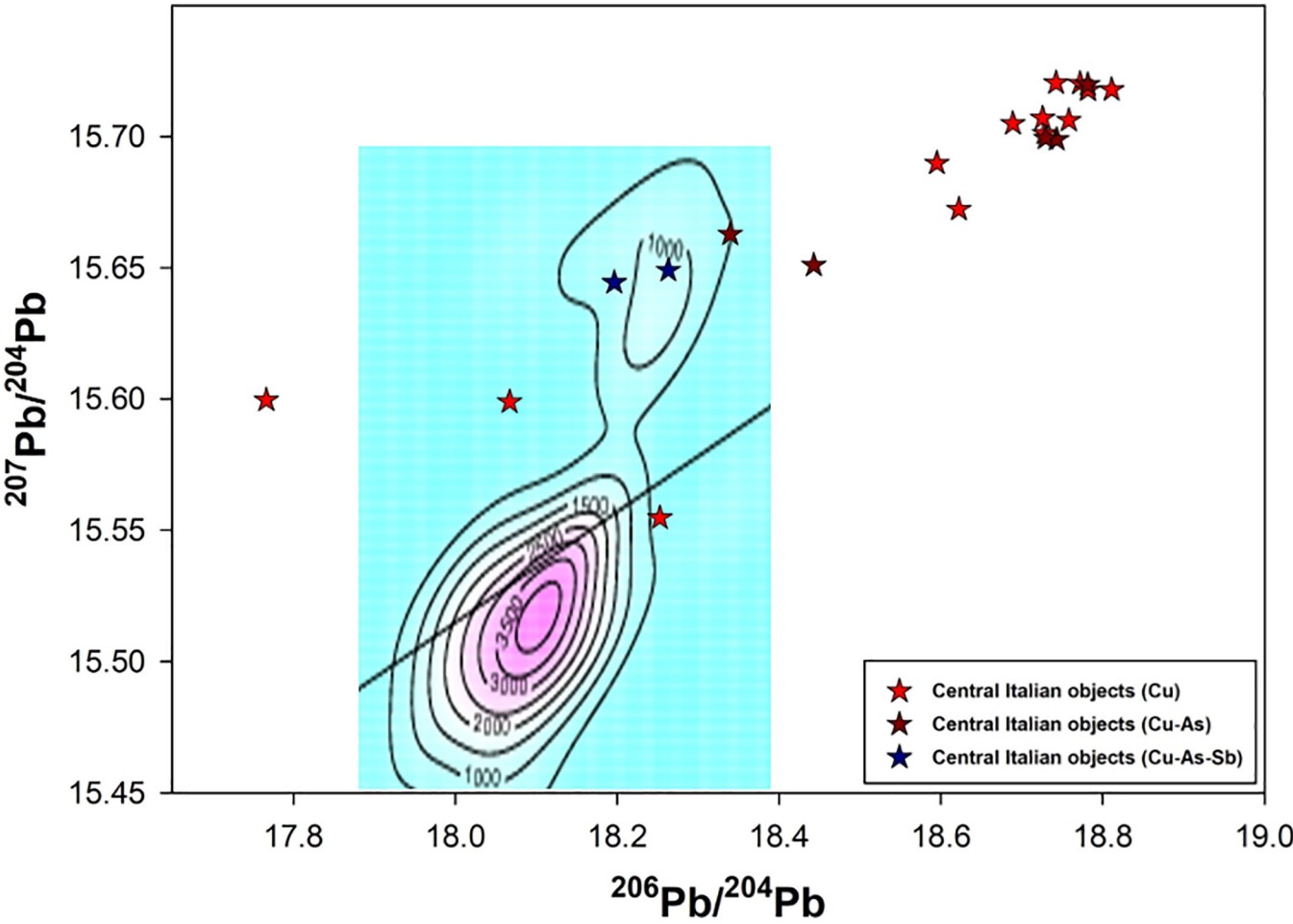

**Fig 10. Comparison of the isotopic signal of central Italian metal objects with the 2D KDE model derived for the Liguria-Apennine mines.**

integrated LIA, chemical characterisation, and archaeological analysis. The research has painted a bold new picture of the dynamics of metalwork procurement and circulation, whose implications go beyond the region examined. Firstly, it has confirmed the importance of the Tuscan Ore Mountains (and neighbouring deposits) as copper-bearing sources; they were exploited from the mid-4[th] millennium BC for the production of most objects in our 20-strong sample. This is an important datum in its own right as virtually no evidence of early workings survives in Tuscany due to extensive historic mining.

Secondly, it has highlighted significant instances of copper procurement from alternative sources. The most likely candidates for this copper metal are the Apuanian Alps (northwest Tuscany), the Western Alps, and–limited to one object–the French Massif Central. Considering object chronology, we have suggested that these deposits, too, might have been exploited from the mid/late 4[th] millennium BC. This is an important finding, which pushes back the beginnings of copper mining in these regions considerably. The discovery of non-Tuscan copper in central Italy at the dawn of the metal age is a significant, and largely unexpected, research result, which warrants further explorations grounded in a larger sample of objects. Further studies should also clarify (a) if the lack of copper metal from Libiola, Monte Loreto, and other ore bodies from the northern Apennines is a visibility issues (due to the small sample size) or a real datum; and (b) if ternary Cu-As-Sb alloys solely characterise copper from the

Western Alps (as the data preliminary indicate) or, as we would tentatively propose, are typical of *both* Western Alpine *and* Tuscan copper (considering the abundant polymetallic ore deposits from the latter region).

Thirdly, the research has highlighted significant evidence of early metal recycling. Considering that most objects in our sample display distinctive regional typological traits, but not all copper was sourced locally, we have suggested that central Italian smiths would have recast foreign-looking objects into familiar ones to satisfy cultural expectations concerning what a metal artefact should look like. Initially, this did not involve any mixing of metal. Old axe-heads were probably cast into new axe-heads, and so were daggers and halberds. Over time, however, the spread of copper recycling as a technological practice would have caused copper from different sources to be mixed together, as shown by the blurred LI signatures of certain $3^{rd}$ millennium BC metals in the sample. This, too, is a significant research result that should be tested in future studies relying on a broader sample size.

Finally, and perhaps most importantly, the research has shown that early metal exchange in Italy and surrounding regions would have followed three networks. The first encompassed the north-eastern Alps, east-central Europe, and the Balkans; the second, the middle and upper Tyrrhenian region, the Western Alps, and the French Midi; and the third, the south-eastern Alps and north-eastern Italy. Central Alpine communities were apparently able to tap into all three circuits. Bolstering and nuancing the results obtained by Perucchetti et al. [23], we have argued that the three circuits showed exceptional vitality and longue durée; in northern Italy, and perhaps further afield, they were still active in the $2^{nd}$ millennium BC.

The broad-brush picture painted in these pages suggests that multiple mechanisms and agents would have underpinned metal exchange within these networks. The objects found near the mining regions might have been procured directly. Those from other areas might have been passed on down the line, until someone decided to commit them to the ground as part of a ritual practice. Other objects yet would have been cast into new objects, the memory of their travels likely erased as the copper metal was melted and poured into a new mould. Whatever circumstances presided to the displacement of individual objects, the research has overall shown that, against a backdrop of regional procurement and circulation, several metals would have entered far-flung exchange networks that, through complex biographies involving new casting, smithing, and shaping events, took them hundreds of miles away from the lands where the copper was first extracted. Ore-rich Tuscany partook in one of these networks. As well as supplying metal to it, its prehistoric communities would have received some of the copper that, whatever its point of origin, constantly circulated through the network. Seen from this perspective, what may first seem to be a curious case of 'bringing coals to Newcastle' turns out to be a natural function of network dynamics, as revealed by integrated archaeology, metal chemistry, and lead isotope analysis.

## Acknowledgments

We are grateful to the following institutions for granting permission to sample and investigate the objects in their care: The British Museum; Museo Archeologico Nazionale di Firenze; Museo Archeologico Nazionale di Siena; Museo Archeologico Nazionale di Arezzo; and the former Soprintendenza per i Beni Archeologici della Toscana. We are especially grateful to Biancamaria Aranguren, Duncan Hook, Fulvia Lo Schiavo, and Ben Roberts for facilitating access to the collections. We thank Igor M. Villa for enabling access to the MC-ICP-MS instrument at the Institut für Geologie, University of Bern; Eva Azzali and Anna Addis for helping with sample preparation and experimental measurements; Cristiano Iaia for advice on object chronology; Marco Benvenuti and Andrea Dini for providing reference materials and

illustrations; Giovanni Carboni for Fig 2; Monica Miari for consenting to reproduce Fig 3; and Eleonora Montanari for formatting the manuscript and images, and preparing Fig 4. We also thank Zofia Anna Stos-Gale and the journal's anonymous reviewers for their helpful comments and constructive criticism. All opinions and errors are our own.

## Author Contributions

**Conceptualization:** Andrea Dolfini, Gilberto Artioli.

**Data curation:** Ivana Angelini, Gilberto Artioli.

**Formal analysis:** Andrea Dolfini, Ivana Angelini.

**Investigation:** Andrea Dolfini, Ivana Angelini.

**Methodology:** Andrea Dolfini, Gilberto Artioli.

**Project administration:** Gilberto Artioli.

**Resources:** Gilberto Artioli.

**Supervision:** Gilberto Artioli.

**Writing – original draft:** Andrea Dolfini, Gilberto Artioli.

**Writing – review & editing:** Andrea Dolfini, Gilberto Artioli.

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
