## [Decision Letter · Decision Letter 0]

11 Sep 2019

PONE-D-19-21830

Copper to Tuscany – Coals to Newcastle? A multidisciplinary investigation of the dynamics of metalwork exchange in prehistoric Italy by lead isotope, chemical, and archaeological analysis

PLOS ONE

Dear Dr Dolfini,

Thank you for submitting your manuscript to PLOS ONE. After careful consideration, we feel that it has merit but does not fully meet PLOS ONE’s publication criteria as it currently stands. Therefore, we invite you to submit a revised version of the manuscript that addresses the points raised during the review process.

We would appreciate receiving your revised manuscript by Oct 26 2019 11:59PM. To enhance the reproducibility of your results, we recommend that if applicable you deposit your laboratory protocols in protocols.io, where a protocol can be assigned its own identifier (DOI) such that it can be cited independently in the future. For instructions see: http://journals.plos.org/plosone/s/submission-guidelines#loc-laboratory-protocols

We look forward to receiving your revised manuscript.

Kind regards,

Peter F. Biehl, PhD

Academic Editor

PLOS ONE

Journal Requirements:

1. In your manuscript, please provide additional information regarding the specimens used in your study. Ensure that you have reported specimen numbers and complete repository information, including museum name and geographic location.

For more information on PLOS ONE's requirements for paleontology and archaeology research, see https://journals.plos.org/plosone/s/submission-guidelines#loc-paleontology-and-archaeology-research.

2. We note that [Figure(s) 1-6] in your submission contain [map/satellite] images which may be copyrighted. All PLOS content is published under the Creative Commons Attribution License (CC BY 4.0), which means that the manuscript, images, and Supporting Information files will be freely available online, and any third party is permitted to access, download, copy, distribute, and use these materials in any way, even commercially, with proper attribution. For these reasons, we cannot publish previously copyrighted maps or satellite images created using proprietary data, such as Google software (Google Maps, Street View, and Earth). For more information, see our copyright guidelines: http://journals.plos.org/plosone/s/licenses-and-copyright.

1.    You may seek permission from the original copyright holder of Figure(s) [1-6] to publish the content specifically under the CC BY 4.0 license. 

Additional Editor Comments (if provided):

Your manuscript has now been seen by three referees, whose comments are appended below. You will see from these comments that while the referees find your work of potential interest, they have raised substantial concerns that must be addressed. In light of these comments, we cannot accept the manuscript for publication, but would be interested in considering a revised version that addresses these serious concerns.

We hope you will find the referees' comments useful as you decide how to proceed. Should presentation of further data and analysis allow you to address these criticisms, we would be happy to look at a substantially revised manuscript. However, please bear in mind that we will be reluctant to approach the referees again in the absence of major revisions.

Reviewers' comments:

Reviewer's Responses to Questions

**Comments to the Author**

1. Is the manuscript technically sound, and do the data support the conclusions?

Reviewer #1: Yes

Reviewer #2: Yes

Reviewer #3: Partly

2. Has the statistical analysis been performed appropriately and rigorously? 

Reviewer #1: Yes

Reviewer #2: Yes

Reviewer #3: N/A

3. Have the authors made all data underlying the findings in their manuscript fully available?

Reviewer #1: Yes

Reviewer #2: Yes

Reviewer #3: Yes

4. Is the manuscript presented in an intelligible fashion and written in standard English?

Reviewer #1: Yes

Reviewer #2: Yes

Reviewer #3: Yes

5. Review Comments to the Author

Reviewer #1: I commend the MS for successfully bringing together archaeometric and archaeological data. Some minor issues that need to be addressed before publication are as follows:

Points for Consideration:

1) The title of the article is different in the MS submission page and in the main body text. Make sure that the MS title is consistent.

2) Lines 63-64: the authors document the ternary alloy of Cu-As-Sb alloy as ‘solely’ in central Italy. It is not clear either the argument is for Italy, European prehistory, or for a broader region? If they also argue for a broader region, existence of Cu-As-Sb alloy is known from Norşuntepe (Anatolia, ca. 3500 BC) and Arslantepe (Anatolia, from levels dated to ca. 3750-3400 and 3400-3000 BC). This point needs clarification.

See relevant bibliography examples:

Palmieri, A., Begemann, F., Schmitt-Strecker, S., Hauptmann, A. 2002. Chemical composition and lead isotopy of metal objects from the “Royal” Tomb and other related finds at Arslantepe, Eastern Anatolia, Paléorient 28: 43-69.

Zwicker, U. 1977. Investigations on extractive metallurgy of Cu/Sb/As ore and excavated smelting products from Norşuntepe (Keban) on the Upper Euphrates (3500-2800 BC), in Aspects of Early Metallurgy, Oddy W.A. (ed.), London. 13-26.

3) For Table 1, please give both 1σ and 2σ probabilities of C14 dates.

4) Line 142 & 738: remove the word ‘negative’ as it does not strengthen the expression in the sentence. This argumentation of the authors also contradict to their writing in lines 484-485, where they clearly use the argument that ‘absence of evidence should not be taken as evidence of absence.’

5) For the chronologies argued between lines 249-263, a recent EU funding is mentioned and a paper presentation in the upcoming EAA is cited. The reviewer, thus could not double-check the chronological data cites in [71].

6) For the objects with suggested provenance from outside Tuscany, antimony and arsenic bearing cupriferous ores are preferred also on the grounds of close proximity. The authors should clarify a) why they have completely ruled out co-smelting of copper ores and arsenic/antimony ores? b) As traces of Sb-containing minerals were detected, and considering the polymetallic nature of the region, could there be another undetected source of Sb available?

7) The assertions in the paragraph (lines 586-596) ground on far too many unconfirmed chronology along with thin artefactual evidence (an axe head). Even though the suggestions are also tempting to push back the date of extractive metallurgy in the French Massif Central, I suggest removal of this part or rephrasing it within the observations presented in the next paragraph (lines: 597-607). My suggestion is also partly because of the PlosOne criteria for publication. The relevant parts in conclusion also need rephrasing in the line of this suggestion.

Reviewer #2: This paper is well written and the description of the results of this research is well substantiated. All references are appropriate and the figures are well presented. Just a few remarks:

1 Line 522: there is a mistake: it should be '... high As, high Sb' not 'As'

2. Lines 694-696: it is surprising that Slovakian Ore Mountains are not mentioned as possible source of copper for the 'central and eastern European communities'. The ores from this location have been widely discussed recently (for exmaple Norgaard et al. 2019) and their exploitation in the 4th-3rd millennium BC have been suggested on several accounts (for example Schreiner 2007, Kowalski et al. 2019).

3. Lines 393-398: The database for ore deposits in Italy sounds very impressive. The OXALID is open access, are there plans to make the database of the Padova University also an open access resource? Such database would be of great value to other researchers.

Reviewer #3: This is an interesting paper that reveals a few important issues related to the early stage of metal working in Italy (and in Europe). This is true, provided that we accept some of the premises made by the authors, namely, that the metal objects that they study are indeed of early Chalcolithic age. I have two major issues with paper and several minor ones.

(1) Most of the objects included in this study came from unsecured locations and do not have precise archaeological context. Their dating to the Chalcolithic period is based on “a recent EU-funded project aiming to date early Italian metals, including those discussed here, by combined morphometric analysis, radiocarbon dating, and a critical review of find contexts [71].” It is important to note; however, that reference #71 is a recent conference abstract. If we accept the author’s claim that “as is typical of Copper Age Italy, they are mostly stray finds, suggesting intentional deposition in the open landscape” the paper can be published. Maybe I am too conservative, but I am used to analyze only archaeological objects that come from a secure, well-constrained archaeological context. I think that the editor should decide whether the use of these objects warrants the publication of the paper.

(2) The paper is too long and too convoluted. Once we accept the dating of the objects, there is one striking result that should be emphasized: 55-75% of metal items were produced from Tuscan copper, either from Southern Tuscany or from the Apuanian Alps, and 25-45% come from outside this region. This is a very important observation that merits the publication of this paper. The rest of the argumentations should be scaled down, and the use of ED models should be mentioned but not presented graphically. I will outline below which of the figures and tables can be omitted, and also comment on other minor aspects of the paper.

Minor comments:

1. The abstract should be more informative, and should include real data.

2. Figure 1 should include a few contemporary cities so readers who are unfortunate to be non-Italian will have an easy way to put the finds in geographical context. Figure 5 should be an insert in Figure 1.

3. Figures 2 and 3 and Tables 1 and 2 are not needed.

4. Line 112: Define LIA here.

5. The meaning “soapstone” for steatite should appear already in line 127.

6. Figure 6: The map should be modified. Replace the words "unit" "province" w/ the rock type/s and geological periods (e.g., Jurassic limestone and dolomite, Tuscan ultramafic 50 Myr pluton/unit).

7. Lines 285 – 301 should be rewritten and included timing (in Myr) of the geological events, as this will be used later in the Pb-Pb plots (see below).

8. Table 4: Should add age of formation to all ores in Myr.

9. The phrasing “The latter two fields have a clear metamorphic and/or hydrothermal character resulting in substantially higher 207Pb/204Pb and 208Pb/204Pb values compared with the Ligurian-Apennine ophiolites, is not correct. There might be other reasons: younger age of formation or higher U and Th concentrations. This issue can be resolved w/ appropriate use of the literature.

10. Figure 7a, b, c should be modified. The authors should add Pb-Pb model age curves. Also, they should verify that there is no mass dependent fractionation (an instrumental artifact) in the database that they use.

11. Lines 383-384: They should report the measured 6/4, 7/4, 8/4 ratios of SRM981 and the errors on each one of them (2�) and the number of independent runs during this study.

12. The authors should emphasize more the issue of mixing, and using the modified Figure 7 they will be better equipped to address this issue.

13. Lines 415-416: “Fig 8. LI signatures of central Italian metal objects (labelled ‘Rinaldone’). 2D projections of the three-dimensional space defined by the measured isotopic ratios 206Pb/204Pb, 207Pb/204Pb, and 208Pb/204Pb.” I do not understand why they needed to label the object “Rinaldone”, and I do not see the 2D projection. All I see are two conventional Pb isotopic plots. In order to simplify this figure, I would downplay the symbols of the ores and add “fields” that mark their typical isotopic values. Also, Figure 8 should be part of the results.

14. Figure 9 is not needed.

15. Consider combining Tables 5 and 6.

16. Lines 508 – 509: In order to test for mixing try plotting 6/4 versus 1/[Pb].

17. Lines 519 – 520, this is part of the most important findings of this paper and should be presented much earlier.

18. Lines 571 – 574 are not needed in light of the what is written in lines 578 – 579.

19. Lines 644 – 645: This is too an important observation and should be mentioned much earlier in the paper, including in the abstract.

20. The use of the term “Oxford group’ method” is not clear, and when I looked at reference 121, the term became even more obscure to me.

21. Figure 10: Use different symbol types (e.g., dots versus stars) for objects and ores.

22. Lines 680 – 708: Too long and too speculative. Make it substantially shorter.

23. Figure 11 is not needed.

24. Conclusions: should be rewritten emphasizing first the important messages of the paper.

25. References: There are a few typos in the references.

6. PLOS authors have the option to publish the peer review history of their article (what does this mean?). If published, this will include your full peer review and any attached files.

Reviewer #1: No

Reviewer #2: Yes: Dr. Zofia Anna Stos-Gale

Reviewer #3: No

---

## [Author Response · Author response to Decision Letter 0]

12 Nov 2019

See 'response to reviewers' file attached to the resubmission

---

## [Decision Letter · Decision Letter 1]

17 Dec 2019

Copper to Tuscany – Coals to Newcastle? The dynamics of metalwork exchange in early Italy

PONE-D-19-21830R1

Dear Dr. Dolfini,

We are pleased to inform you that your manuscript has been judged scientifically suitable for publication and will be formally accepted for publication once it complies with all outstanding technical requirements.

With kind regards,

Peter F. Biehl, PhD

Academic Editor

PLOS ONE

Note from Journal Editoral Staff:

Note that while Reviewer #3 makes some additional suggestions regarding the presentation of the work, the Academic Editor does not feel that these changes are required for the manuscript to meet PLOS ONE's publication criteria and that the manuscript can be published as written.

Reviewers' comments:

Reviewer's Responses to Questions

**Comments to the Author**

1. If the authors have adequately addressed your comments raised in a previous round of review and you feel that this manuscript is now acceptable for publication, you may indicate that here to bypass the “Comments to the Author” section, enter your conflict of interest statement in the “Confidential to Editor” section, and submit your "Accept" recommendation.

Reviewer #1: All comments have been addressed

Reviewer #2: All comments have been addressed

Reviewer #3: (No Response)

2. Is the manuscript technically sound, and do the data support the conclusions?

Reviewer #1: Yes

Reviewer #2: Yes

Reviewer #3: Yes

3. Has the statistical analysis been performed appropriately and rigorously? 

Reviewer #1: Yes

Reviewer #2: Yes

Reviewer #3: Yes

4. Have the authors made all data underlying the findings in their manuscript fully available?

Reviewer #1: Yes

Reviewer #2: Yes

Reviewer #3: Yes

5. Is the manuscript presented in an intelligible fashion and written in standard English?

Reviewer #1: Yes

Reviewer #2: Yes

Reviewer #3: No

6. Review Comments to the Author

Reviewer #1: (No Response)

Reviewer #2: I agree with all answers to the referees and corrections. I think that this paper now is ready for publication.

Reviewer #3: 1. The paper is too long.

2. The abstract should be more informative, and should include real data.

3. Figures 2 and 3 and Tables 1 and 2 are not needed.

4. The “Cultural context” section is too long

5. The description of the ores is too long.

6. Table 4 & Figs. 7, 8 should be part of the results and discussion section.

7. Section “Ore geology and lead isotope fingerprinting” should come before the “Methods” and should be part of the “Introduction” along w/ Figure 6 and Table 3.

8. The discussion is too long and too speculative.

7. PLOS authors have the option to publish the peer review history of their article (what does this mean?). If published, this will include your full peer review and any attached files.

Reviewer #1: No

Reviewer #2: Yes: Zofia Anna Stos-Gale

Reviewer #3: No

---

## [Editor Report · Acceptance letter]

30 Dec 2019

PONE-D-19-21830R1 

Copper to Tuscany – Coals to Newcastle? The dynamics of metalwork exchange in early Italy 

Dear Dr. Dolfini:

I am pleased to inform you that your manuscript has been deemed suitable for publication in PLOS ONE. Congratulations! Your manuscript is now with our production department. 

With kind regards,

on behalf of

Dr. Peter F. Biehl 

Academic Editor

PLOS ONE